# An extracellular vesicle targeting ligand that binds to Arc proteins and facilitates Arc transport in vivo

Peter H Lee*, Michael Anaya, Mark S Ladinsky, Justin M Reitsma†, Kai Zinn*

Division of Biology and Biological Engineering, California Institute of Technology, Pasadena, United States

**Abstract** Communication between distant cells can be mediated by extracellular vesicles (EVs) that deliver proteins and RNAs to recipient cells. Little is known about how EVs are targeted to specific cell types. Here, we identify the *Drosophila* cell-surface protein Stranded at second (Sas) as a targeting ligand for EVs. Full-length Sas is present in EV preparations from transfected *Drosophila* Schneider 2 (S2) cells. Sas is a binding partner for the Ptp10D receptor tyrosine phosphatase, and Sas-bearing EVs preferentially target to cells expressing Ptp10D. We used co-immunoprecipitation and peptide binding to show that the cytoplasmic domain (ICD) of Sas binds to dArc1 and mammalian Arc. dArc1 and Arc are related to retrotransposon Gag proteins. They form virus-like capsids which encapsulate *Arc* and other mRNAs and are transported between cells via EVs. The Sas ICD contains a motif required for dArc1 binding that is shared by the mammalian and *Drosophila* amyloid precursor protein (APP) orthologs, and the APP ICD also binds to mammalian Arc. Sas facilitates delivery of dArc1 capsids bearing *dArc1* mRNA into distant Ptp10D-expressing recipient cells in vivo.

**\*For correspondence:**
hlee@caltech.edu (PHL);
zinnk@caltech.edu (KZ)

**Present address:** †AbbVie, Illinois, United States

## Editor's evaluation

The manuscript addresses how extracellular vesicles (EV) are targeted to their recipient cells once they are produced and released. The study shows that a transmembrane protein Sas gets incorporated into EVs, and this protein binds to its receptor Ptp10D on target cells, thus targeting the EVs. The expression of dARC1 in the EV-producing cells leads to the increased expression of the protein dARC1 protein and mRNAs in the recipient cells.

## Introduction

Extracellular vesicles (EVs) are mediators of cell-cell communication that transport specific protein and RNA cargoes. They are a heterogeneous collection of vesicular structures that are exported from cells by a variety of mechanisms. Exosomes are 30–150 nm in diameter and are released into cell supernatants *via* fusion of multivesicular bodies (MVBs) with the plasma membrane. Exosomes and other EVs carry specific proteins and RNAs, and EVs derived from different cell types contain different cargoes. EV cargoes are biomarkers for specific diseases. Because EVs can encapsulate RNAs and protect them from degradation, and then deliver those RNAs to recipient cells, they represent a promising new type of therapeutic agent (*O'Brien et al., 2020*; *Teng and Fussenegger, 2020*).

Little is known about mechanisms involved in EV targeting to specific cell types. EVs are internalized into cells after receptor binding using a variety of endocytic mechanisms, resulting in the delivery of their cargoes into the recipient cells. They can also directly activate intracellular signaling without endocytosis by interacting with cell surface receptors. In this paper, we identify Stranded at second

(Sas), a large *Drosophila* cell surface protein (CSP; *Schonbaum et al., 1992*), as an EV targeting ligand. Full-length Sas (Sas^FL) has an extracellular domain (ECD) containing a signal peptide, a unique N-terminal region, four von Willebrand factor C (VWFC) domains, and three Fibronectin Type III (FN-III) repeats (*Figure 1a*). A shorter isoform, Sas^short, which is translated from an alternatively spliced mRNA, lacks a 345 amino acid (aa) region of the ECD. The two Sas isoforms share a single transmembrane (TM) domain and a short cytoplasmic domain (ICD).

Sas is commonly used as a marker for the apical surfaces of epithelially derived cells, including tracheal cells in the respiratory system. *sas* mutant larvae die at or before second instar (hence the name stranded at second, which is derived from baseball terminology) and have tracheal phenotypes in larvae but no known phenotypes in embryos (*Schonbaum et al., 1992*). A tyrosine motif in the Sas ICD was found to bind to the PTB domain of Numb (*Chien et al., 1998*), an protein involved in endocytosis that is a negative regulator of Notch signaling. Sas has no obvious mammalian orthologs, but there are many mammalian CSPs that contain VWFC and FN-III domains.

In earlier work, we identified the receptor tyrosine phosphatase (RPTP) Ptp10D as a binding partner for Sas, and showed that Sas::Ptp10D interactions regulate embryonic axon guidance, as well as glial migration and proliferation (*Lee et al., 2013*). Ptp10D is one of the two *Drosophila* R3 subfamily RPTPs, which have ECDs composed of long chains of FN-III repeats. Sas::Ptp10D interactions also control the elimination of neoplastic epithelial clones by surrounding normal tissue. Sas is on normal epithelial cells, and it relocalizes to the parts of their cell surfaces that are adjacent to the neoplastic clone and binds to Ptp10D on the neoplastic cells. Ptp10D in turn relocalizes and dephosphorylates the EGF receptor tyrosine kinase, leading to death of the neoplastic cells (*Yamamoto et al., 2017*). The Sas ECD has other binding partners as well, because it interacts with cells that do not express Ptp10D in live embryo staining assays (*Lee et al., 2013*).

Mammalian Arc is a locally translated dendritic protein that regulates synaptic plasticity, in part by modulating endocytosis of AMPA receptors (*Chowdhury et al., 2006*; *Shepherd et al., 2006*). The *Drosophila* genome encodes two proteins distantly related to mammalian Arc, dArc1 and dArc2. The *dArc2* gene, which encodes a truncated protein, was likely generated by a gene duplication, and the *dArc1* and *dArc2* genes are adjacent (*Mattaliano et al., 2007*). dArc1 functions in larval and adult brain neurons to regulate aspects of metabolism (*Mattaliano et al., 2007*; *Mosher et al., 2015*; *Keith et al., 2021*). Arc and dArc1 evolved independently from retrotransposon Gag proteins (*Shepherd, 2018*). Remarkably, they both form virus-like capsids that can encapsulate *Arc* mRNAs and are transported between cells via EVs (*Ashley et al., 2018*; *Pastuzyn et al., 2018*; *Hantak et al., 2021*). dArc1, but not dArc2, has a C-terminal $Zn^{2+}$ finger that might be involved in nucleic acid binding (*Pastuzyn et al., 2018*; *Erlendsson et al., 2020*). Mammalian Arc lacks $Zn^{2+}$ fingers, but RNA is required for normal capsid assembly (*Pastuzyn et al., 2018*). *Drosophila* dArc1 capsids bearing *dArc1* mRNA move from neurons to muscles across larval neuromuscular junction (NMJ) synapses, and dArc1 transfer is required for activity-induced induction of morphological synaptic plasticity (*Ashley et al., 2018*).

Here we show that full-length Sas expressed in cultured *Drosophila* cells localizes to EVs that preferentially target to cells expressing Ptp10D. Sas binds to dArc1 in EVs via a tyrosine motif in its ICD that is conserved between Sas and the human and *Drosophila* APP orthologs. This finding led us to examine mammalian Arc as well, and we showed that it also binds to Sas and APP. The interaction between APP and Arc is of interest because several studies have implicated Arc in control of β-amyloid accumulation and Alzheimer's disease (AD) (*Wu et al., 2011*; *Landgren et al., 2012*; *Bi et al., 2018*). APP also localizes to EVs (*Laulagnier et al., 2018*; *Pérez-González et al., 2020*).

Since Sas binds to dArc1, which can encapsulate its own mRNA (*Ashley et al., 2018*), we then investigated whether Sas EVs can target *dArc1* mRNA to Ptp10D-expressing recipient cells in vivo. Here, we show that co-expression of dArc1 and full-length Sas in embryonic salivary glands (SGs) causes *dArc1* mRNA to appear in distant tracheal cells that express Ptp10D.

## Results and discussion
### Sas is an EV targeting ligand

Sas has two isoforms, translated from alternatively spliced mRNAs. Sas^FL (PB/PD isoform) is a 1693 aa single-transmembrane cell-surface protein with a short (37 aa) ICD. It contains a 345 aa region (EVT)

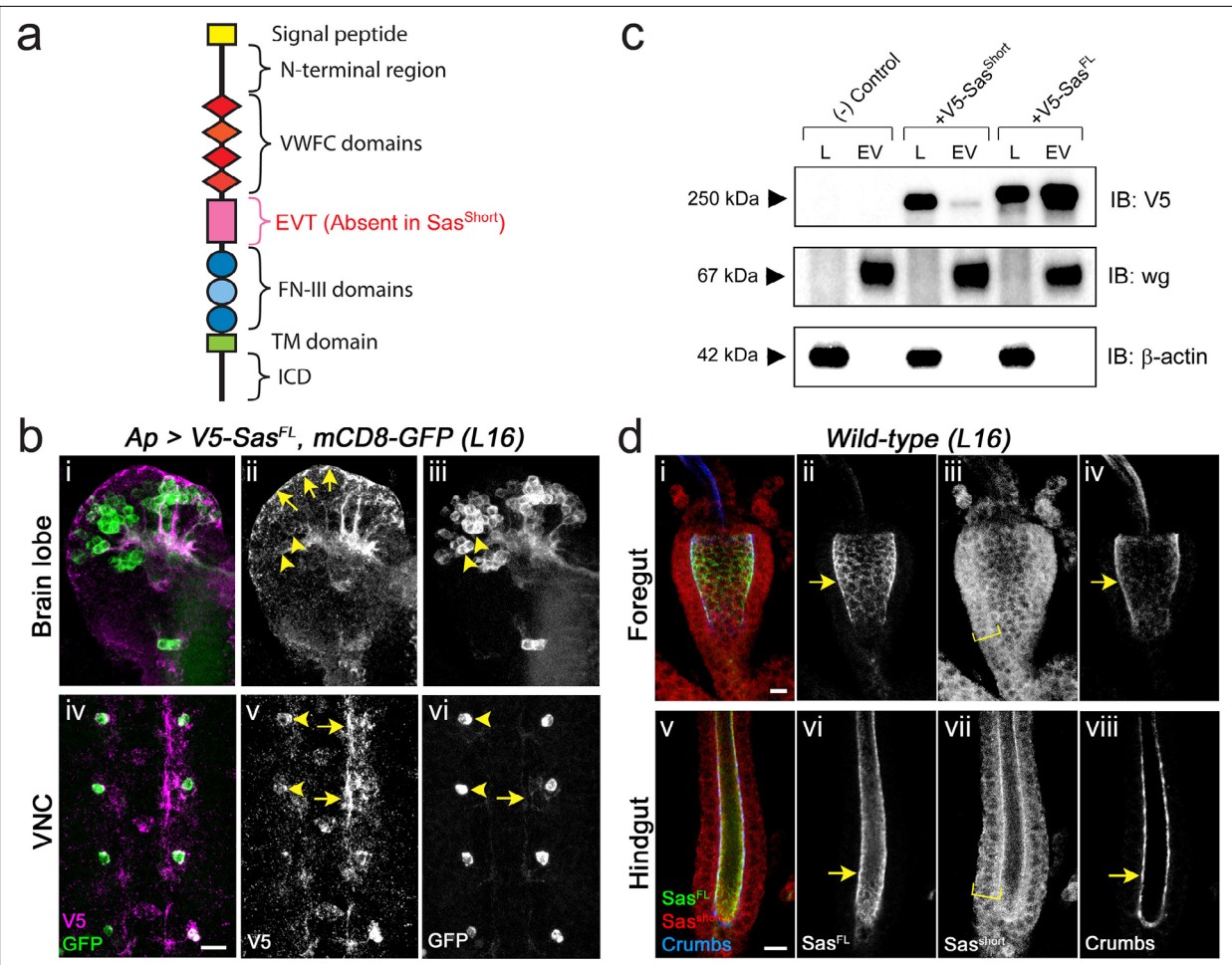

**Figure 1.** Localization of Sas isoforms. (**a**) Schematic diagram of the Sas$^{FL}$ protein. The Sas$^{short}$ isoform lacks the EVT region. (**b**) Sas$^{FL}$ moves away from expressing cells. V5-Sas$^{FL}$ was expressed together with mCD8-GFP (transmembrane CSP) in Apterous neurons in late stage 16 embryos. (**i,iv**) double-labeling (V5, magenta; GFP, green); (**ii, v**) V5 channel; (**iii, vi**) GFP channel. (**i-iii**) Brain lobes. GFP labels cell bodies (arrowheads in **iii**) and axon tracts. V5 labels cell bodies only weakly (arrowheads in **ii**), strongly labels some axon tracts, and localizes to the periphery (sheath) of the brain lobes (arrows in **ii**). (**iv-vi**) Ventral nerve cord. GFP strongly labels Ap VNC cell bodies (arrowheads in **vi**) and weakly labels Ap axons (arrow in **vi**). V5 weakly labels cell bodies (arrowheads in **v**), and strongly labels segments of axons (arrows in **v**) and areas adjacent to the axons. These may be extracellular matrix and/or glial sheaths. Scale bar, 10 μm. (**c**) Western blot, showing that Sas$^{FL}$ localizes to EVs. EVs were prepared from supernatants from S2 cells expressing the indicated protein, and equal amounts of cell lysate proteins and EV proteins were loaded on the gel. Top panel, anti-V5 blot. Sas$^{FL}$ migrates slightly above the 250 kD marker, and Sas$^{short}$ slightly below it. Middle panel, Wg (EV marker). Almost all of the Wg is in EVs. Bottom panel, anti-β-actin (cytoplasmic marker) blot. There is no signal in EVs. In these samples, 60% of Sas$^{FL}$ and 10% of Sas$^{short}$ is in EVs; the average over 4 experiments was 46% in EVs for Sas$^{FL}$ and 10% for Sas$^{short}$. The absence of β-actin signal in the EVs shows that they are not heavily contaminated by cytosol. (**d**) Localization of endogenous Sas isoforms in the embryonic gut. *Wild-type* (*wt*) late stage 16 embryos were triple-stained for Sas$^{FL}$ (using the [*Schonbaum et al., 1992*] antiserum, which primarily recognizes the EVT region; green), Sas$^{short}$ (using our antipeptide antibody; red), and Crumbs (apical marker; blue). (**i-iv**) Foregut; (**v-viii**), hindgut. Note that in both gut regions Sas$^{FL}$ colocalizes with Crumbs at the apical (luminal) cell surfaces (arrows), while anti-Sas$^{short}$ labels the entire width of the gut wall (brackets). See *Figure 1—figure supplement 1* for images of anti-Sas$^{short}$ staining of *wt* and *sas* mutant embryos, demonstrating antibody specificity. Scale bar in **d-i**, 20 μm; in (**d-v**), 10 μm.

The online version of this article includes the following source data and figure supplement(s) for figure 1:

**Source data 1.** Source data files for *Figure 1* include raw and labelled images for the western blots shown in panel (c); source data files for *Figure 1— figure supplement 1* include raw and labelled images for the western blots shown in panels (a) and (d).

**Figure supplement 1.** Recognition of Sas isoforms by anti-Sas antibodies, retention of Sas$^{short}$ by expressing cells, and comparison of EVs generated by ultracentrifugation and chemical reagents (Exosome Isolation Kit).

**Figure supplement 1—source data 1.** Raw data from western blots comparing two EV samples.

between the VWFC and FN-III domains that is lacking in the Sas<sup>short</sup> (PA/PC) isoform (*Figure 1a*). We expressed Sas<sup>FL</sup> tagged with an N-terminal V5 epitope tag (inserted immediately after the signal sequence) in embryonic late stage 16 Apterous (Ap) neurons, which consist of paired neurons (one per hemisegment) in the ventral nerve cord (VNC) and scattered neurons in the brain lobes. It was expressed together with mCD8-GFP, which is also a transmembrane CSP. The GFP signal was restricted to Ap neuron cell bodies, with faint staining on the axons. However, V5-Sas<sup>FL</sup> was observed in sheaths around brain lobes and in areas adjacent to axons in the VNC, as well as in puncta throughout the VNC and brain (*Figure 1b*). The V5 signal in cell bodies was very weak, especially in the brain.

We also expressed V5-Sas<sup>short</sup> in Ap neurons, and observed that V5 staining was restricted to cell bodies and axons in the VNC, and to cell bodies in the brain (*Figure 1—figure supplement 1*). Thus, although Sas<sup>FL</sup> and Sas<sup>short</sup> have the same TM and ICD, they differ in subcellular localization, with Sas<sup>short</sup> being retained in expressing cells and Sas<sup>FL</sup> moving away from these cells and into the extracellular matrix.

Movement of V5-Sas<sup>FL</sup>, and presumably of endogenous Sas<sup>FL</sup>, away from its source could occur through cleavage of the Sas ECD from the cell surface or by release of intact Sas in EVs. To distinguish between these possibilities, we expressed V5-tagged Sas<sup>FL</sup> and Sas<sup>short</sup> in transiently transfected *Drosophila* Schneider 2 (S2) cells, which express endogenous Sas at almost undetectable levels. We prepared EVs from S2 cell supernatants using the Invitrogen Exosome Isolation Kit or by ultracentrifugation, and analyzed their contents by western blotting. EVs contained the Wg protein, which is a marker for EVs in S2 cells and is present at very low levels in lysates (*Koles et al., 2012*; *Figure 1c*, *Figure 1—figure supplement 1*). They lacked β-actin, showing that they are not heavily contaminated by cytosol. We found that about 50% of V5-Sas<sup>FL</sup> localized to EVs, while ~90% of V5-Sas<sup>short</sup> was retained in the cell lysate (*Figure 1c*). We did not observe any proteolytic cleavage products in EVs or unpurified supernatants.

EV preparations made using the Exosome Isolation kit have similar characteristics to those generated by ultracentrifugation (*Skottvoll et al., 2019*), and the kit requires much less material, allowing EVs to be purified from small-scale transfections. However, to confirm results obtained with the kit, we also purified EVs from S2 cells expressing V5-tagged Sas<sup>FL</sup> using a standard ultracentrifugation protocol (*Théry et al., 2006*), and showed that Sas<sup>FL</sup> and Wg are also present in these EVs (*Figure 1—figure supplement 1*).

The commonly used rabbit antiserum against Sas primarily recognizes the EVT region, so embryo staining reveals the localization of endogenous Sas<sup>FL</sup> (*Schonbaum et al., 1992*). To visualize Sas<sup>short</sup>, we made an anti-peptide antibody against a sequence spanning an exon junction in the PA/PC isoforms. This recognizes Sas<sup>short</sup>, and staining with the antibody is eliminated in *sas* mutant embryos (*Figure 1—figure supplement 1*). Double-staining of the foregut and hindgut with the two Sas antibodies showed that Sas<sup>FL</sup> localizes to apical cell surfaces, while Sas<sup>short</sup> is distributed across the entire cell membrane (*Figure 1d*). These data imply that the EVT sequence lacking in Sas<sup>short</sup> is required for both apical localization and targeting to EVs. Polarized cells can release EVs with different cargoes from their apical and basolateral surfaces (*Matsui et al., 2021*), so EV targeting could be downstream of apical localization in vivo. S2 cells are unpolarized, however, so this mechanism is unlikely to apply to trafficking of Sas<sup>FL</sup> to EVs in cultured S2s.

## Analysis of Sas<sup>FL</sup> EVs by electron microscopy

To demonstrate that Sas is actually on EVs, we used immuno-EM and EM tomography to analyze purified EV preparations from V5-Sas<sup>FL</sup>-expressing S2 cells. The tomographic images showed that the EVs span a range of sizes, from ~30 nm in diameter to >100 nm, and that they are a mixture of single and double-membrane vesicles (*Figure 2c and d*, *Figure 2—figure supplement 1*). For immuno-EM, we incubated EVs with anti-V5, followed by gold-labeled anti-mouse secondary antibody. *Figure 2a* shows a typical image, in which an EV is associated with multiple 10 nm gold particles. The distance between the EV membrane (yellow bracket: diameter of the vesicle) and a gold particle (white bracket: distance between membrane and a particle) can be more than 40 nm. This likely reflects the large size of the Sas ECD, in which the N-terminal V5 epitope is separated by 1590 aa from the TM domain. The distance is variable, however, because the Sas ECD, which is composed of a chain of domains separated by linkers, is likely to be flexible and able to adopt many different conformations. The region outside the membrane boundary is of higher density, probably because it represents the

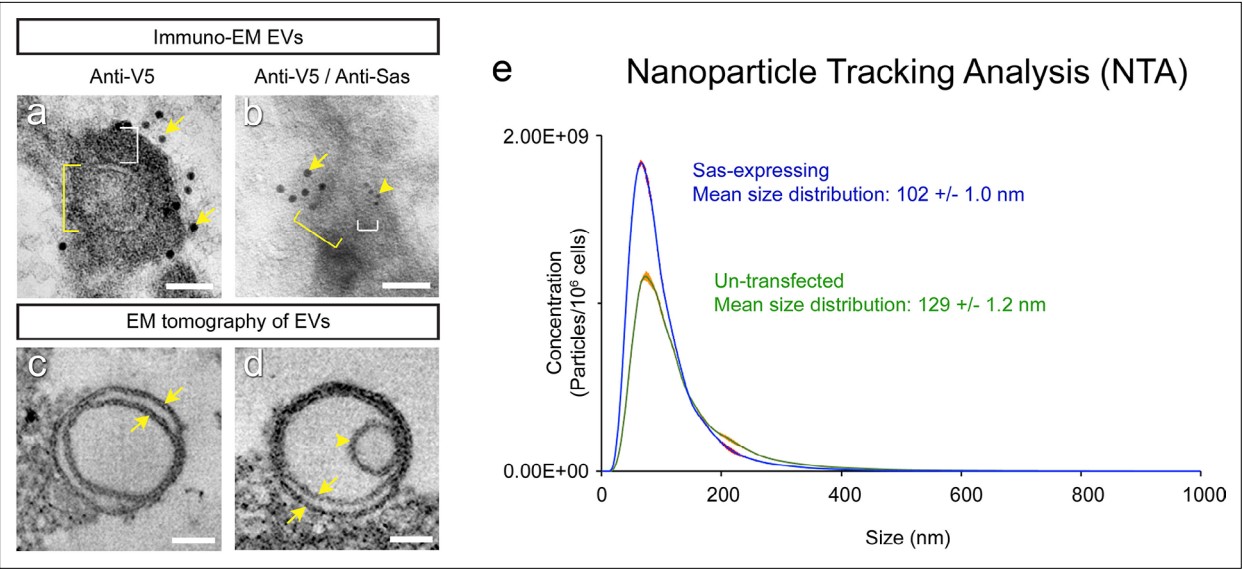

**Figure 2.** Analysis of EVs by electron microscopy and nanoparticle tracking analysis. (**a, b**) Immuno-EM images of EVs from a purified EV prep from V5-Sas[FL]-expressing S2 cells. EV outline (membrane) diameters are indicated by yellow brackets. White brackets, separation between EV outline and a gold particle. (**a**) Immuno-EM with 10 nm anti-V5 gold particles (arrows). (**b**) Immuno-EM with both 10 nm anti-V5 (large gold, arrow) and 5 nm anti-Sas (small gold, arrowhead). (**c**) EM tomogram of an empty double-membrane vesicle (arrows). Apparent EV sizes differ between immuno-EM and tomography, which use very different preparation methods. A low-mag view of a single slice from an EM tomogram of an EV preparation is shown in *Figure 2—figure supplement 1*. (**d**) EM tomogram of a double-membrane vesicle (arrows) with a capsid-sized denser object inside it (arrowhead). *Figure 2—figure supplement 1* shows a 3D reconstruction of this EV. Empty and filled single-membrane EVs were also observed (*Figure 2—figure supplement 1*). Scale bars in (**a–d**), 50 nm. (**e**) Nanoparticle Tracking Analysis (conventional NTA with NanoSight) of purified EV preparations from untransfected (green curve) and Sas[FL]-expressing (blue curve) S2 cells. The mean size distribution is indicated. Standard error indicated by red color around curves. Source data files include an Excel file of raw data for the NTA analysis, the conversion of the numbers from numbers of EVs per sample to numbers of EVs per cell, based on cell counts, and plots of the data.

The online version of this article includes the following video, source data, and figure supplement(s) for figure 2:

**Source data 1.** NTA raw data.

**Figure supplement 1.** EM analysis of EVs from Sas[FL]-expressing S2 cells.

**Figure 2—video 1.** This video shows the 3D reconstruction of the tomogram of the EV shown in *Figure 2d*, which contains a denser object within it. https://elifesciences.org/articles/82874/figures#fig2video1

protein sheath around the EV membrane. To further characterize Sas localization, we performed an experiment in which EVs were incubated with both mouse anti-V5 and rabbit anti-Sas, which primarily recognizes the EVT region in the middle of the ECD, followed by 10 nm gold particle-labeled anti-mouse secondary antibody and 5 nm gold particle-labeled anti-rabbit secondary antibody. *Figure 2b* shows an EV that is associated with multiple 10 nm (arrow) and 5 nm (arrowhead) gold particles.

To analyze the numbers and sizes of EVs from Sas[FL]-expressing and control S2s, we examined purified EVs using Nanoparticle Tracking Analysis with a NanoSight instrument (NTA, System Biosciences, LLC). We observed that the distribution of EV diameters was shifted toward smaller values in cells expressing Sas[FL] (mean diameter = 102 nm, *vs*. 129 nm for control cells) (*Figure 2e*). Expression of Sas[FL] increased the number of EVs per cell in the exosome size range (30–150 nm in diameter) by 44%, and the number of EVs per cell of <100 nm in diameter by 72%, suggesting that the presence of high levels of Sas[FL] increases the rate of EV production.

Many of the EVs from V5-Sas[FL]-expressing S2 cells EVs had denser objects within their boundaries (*Figure 2d*, *Figure 2—figure supplement 1*). These were typically 30–40 nm in diameter. Such objects were also found within EVs from untransfected S2 cells (*Ashley et al., 2018*), so their presence does not require expression of Sas[FL].

## Sas[FL] EVs target to cells expressing Ptp10D

Having shown that Sas[FL] moves away from expressing neurons in the embryo and is an EV component, we then asked whether it can be incorporated into distant cells in vivo, presumably through

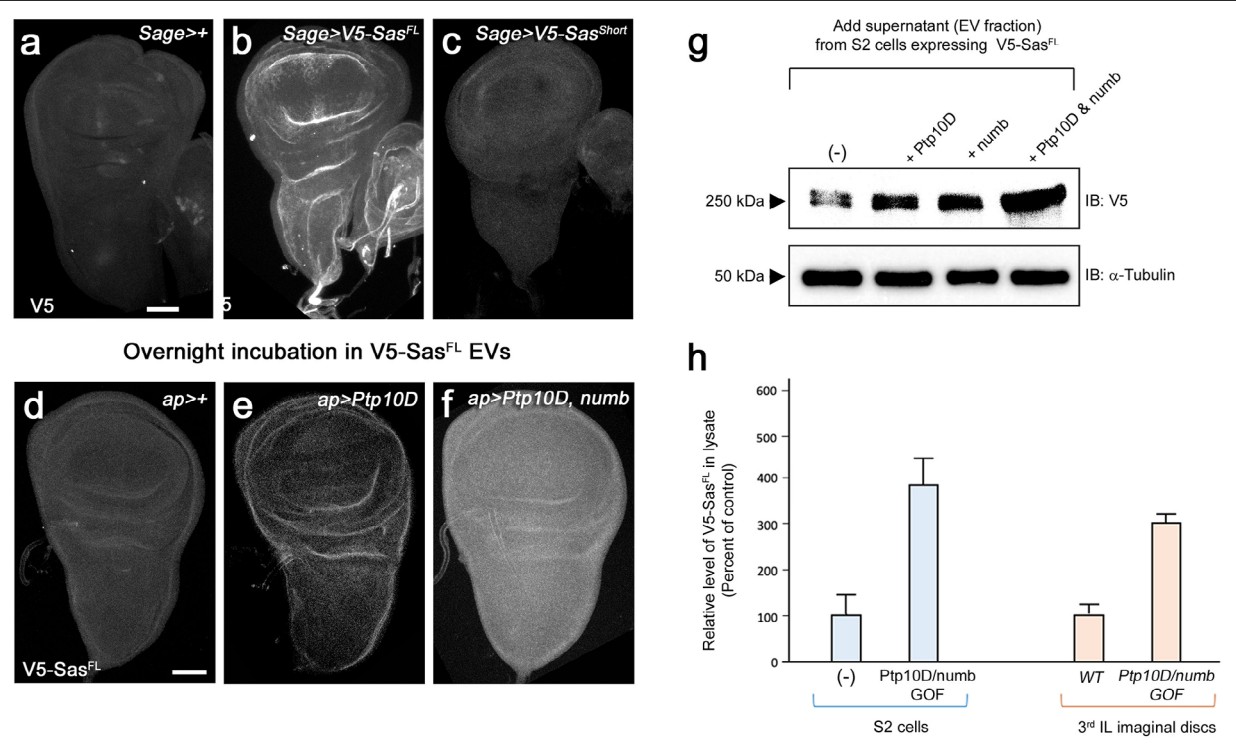

**Figure 3.** Transfer of Sas^FL to recipient cells. (**a**) A third instar wing disc, with a portion of the haltere disc (right side), from a *Sage-GAL4/+* (SG-specific driver) larva, showing no V5 staining. (**b**) A wing disc, with a portion of the haltere disc, from a *Sage >V5-Sas^FL* larva, showing bright V5 staining. (**c**) A wing disc, with a portion of the haltere disc, from a *Sage >V5-Sas^short* larva, showing little or no V5 staining. Imaginal discs display no expression of GFP or mCherry reporters driven by *Sage-GAL4*. (**d-f**) Wing discs incubated with EVs from V5-Sas^FL-expressing S2 cells and stained with anti-V5. For anti-Ptp10D and anti-Numb staining, see *Figure 3—figure supplement 1*. (**d**) *ap-GAL4/+*; (**e**) *ap >Ptp10 D*; (**f**) *ap >Ptp10 D+Numb*. (**d**) Low levels of anti-V5 staining are observed. (**e**) Higher levels are observed in disc folds, which also express Ptp10D (*Figure 3—figure supplement 1*). (**f**) Bright anti-V5 staining is observed throughout the disc. This pattern matches anti-Numb staining (*Figure 3—figure supplement 1*). Scale bars in (**a**) and (**d**), 50 μm. (**g**) Transfer of Sas^FL from EVs into recipient S2 cells. Supernatants from S2 cells expressing V5-Sas^FL were incubated with cultures of untransfected S2 cells or cells expressing Ptp10D, Numb, or both, and cell lysates analyzed by western blotting with anti-V5. Note that V5-Sas^FL levels were elevated relative to control cells by expression of either Ptp10D or Numb, and that levels were further increased by coexpression of Ptp10D and Numb coexpression. (**h**) Quantitation of results from panels (**d-f**) and (**g**). Levels of transferred V5-Sas^FL were increased by ~fourfold relative to untransfected controls by Ptp10D+Numb coexpression in S2 cells (n=6), and by ~threefold relative to ap-GAL4/+control by Ptp10D+Numb coexpression in wing discs (n=5). Quantitation was done using Image J.

The online version of this article includes the following source data and figure supplement(s) for figure 3:

**Source data 1.** Source data files include raw and labelled images for the western blots shown in panel (**g**), and an Excel file of the quantitation of the western blot and disc immunofluorescence signals used to generate panel (**h**).

**Figure supplement 1.** Visualization of all three channels for the triple-stained wing discs shown in *Figure 3*, and expression of the Sage-GAL4 driver in SGs in whole larvae.

endocytosis of EVs. We expressed V5-Sas^FL in 3^rd instar larval salivary glands (SGs) using an SG-specific GAL4 driver, *Sage-GAL4*. SG-specific expression of dsRed from this driver in whole larvae is shown in *Figure 3—figure supplement 1*. We then visualized V5 staining in other tissues. We found that V5-Sas^FL made in SGs is present in imaginal discs, which are separated from SGs by larval hemo-lymph (wing and haltere discs shown in *Figure 3a–c*). There was no V5 staining in imaginal discs from driver-alone larvae (compare *Figure 3a and b*). We also expressed V5-Sas^short in SGs, and this was not observed in imaginal discs (*Figure 3c*). This is consistent with the observation that V5-Sas^short does not move away from expressing Ap neurons and is restricted to cell lysates in S2 cells (*Figure 1b and c*). *wt* discs express the Sas receptor Ptp10D at low levels (see *Figure 3—figure supplement 1*), but we do not know whether Ptp10D is required for Sas^FL binding to discs. Overexpressed Ptp10D can stimulate Sas^FL accumulation in discs, however (see *Figure 3—figure supplement 1*).

To examine the mechanisms involved in specific targeting of Sas EVs, we added supernatants (EV fraction) from V5-Sas$^{FL}$-expressing S2 cells to S2 cell cultures and analyzed recipient cell lysates by western blotting. We observed that expression of the Sas receptor Ptp10D in recipient cells increased V5-Sas$^{FL}$ levels in these cells, as did expression of Numb, a regulator of endocytosis that binds to the Sas ICD (*Chien et al., 1998*). Expression of both Ptp10D and Numb produced a synergistic effect, increasing V5-Sas$^{FL}$ by ~fourfold relative to untransfected recipient cells (*Figure 3g–h*). We speculate that binding of Numb to the Sas ICD increases Sas uptake and/or protects endocytosed Sas from degradation.

We then developed an assay to examine the effects of Ptp10D and Numb on Sas targeting in larval cells by incubating dissected 3$^{rd}$ instar wing imaginal discs with V5-Sas$^{FL}$ supernatants. Wing discs from driver-alone (control) larvae displayed weak V5 staining after incubation with V5-Sas$^{FL}$ EVs. Staining was increased in discs from larvae expressing Ptp10D in imaginal discs, and further elevated (~three-fold increase relative to driver control) by expression of both Ptp10D and Numb (*Figure 3d–f and h*; *Figure 3—figure supplement 1*).

## Sas binds to dArc1 and mammalian Arc via a conserved tyrosine motif

To examine whether Sas interacts with specific EV cargoes, we made EV preparations from S2 cells expressing V5-Sas$^{FL}$ and from untransfected control cells, lysed them with nonionic detergent, incubated the lysate with anti-V5-coupled magnetic beads, and analyzed bead-bound proteins by mass spectrometry (*Figure 4a*, *Supplementary file 1*). We ranked the identified proteins by their degree of enrichment in the V5-Sas$^{FL}$ sample relative to the control. Proteins that are present in the V5-Sas$^{FL}$ sample should include EV cargoes that bind to Sas$^{FL}$ and are therefore present in V5 IPs, plus proteins that nonspecifically bind to V5 beads. Proteins in the control sample would be only those that nonspecifically bind to V5 beads. We observed that the most highly enriched protein (after Sas itself) is dArc1 (22-fold) (*Figure 4b*). dArc2 is #7 on the list (6-fold). *dArc1* mRNA, presumably encapsulated within dArc1 capsids, is known to be a prominent mRNA component of EVs from *Drosophila* cultured cells (*Lefebvre et al., 2016*; *Ashley et al., 2018*). Our data suggest that dArc1, and possibly dArc2, associate with Sas$^{FL}$, since they are enriched in V5 IPs from cells expressing V5-Sas$^{FL}$. We then went on to show that dArc1 binds directly to the Sas ICD (see below).

The presence of Arc proteins in EVs is consistent with published data from both mammalian and *Drosophila* cell cultures. EVs from media of short-term cultures of mouse cortical neurons were shown to contain denser objects whose size (~30 nm in diameter) was consistent with mammalian Arc capsids, and which were associated with anti-Arc gold particles (*Pastuzyn et al., 2018*). For dArc1, capsid-like structures that bound to anti-dArc1 gold particles were detected in lysed preparations of EVs from S2 cells (*Ashley et al., 2018*). The dArc1 capsid is 37 nm in diameter (*Erlendsson et al., 2020*; *Hallin et al., 2021*). As described above, EM tomography of EVs from Sas$^{FL}$-expressing S2 cells showed denser objects of 30–40 nm in diameter within many of them (*Figure 2d*, *Figure 2—figure supplement 1*), consistent with the idea that these are dArc1 capsids. A video of a 3D reconstruction of the EM tomogram of the EV in *Figure 2d* is included, and *Figure 2—figure supplement 1* shows a still image from this video.

Other proteins observed in the mass spectrometric analysis that were present at higher levels in the IP from V5-Sas$^{FL}$-expressing EVs included small ribonucleoproteins (SmE and SmF), a ribosomal protein (NHP2), and a collagen (Vkg; *Figure 4b*). Proteins in these categories were found to be major EV components in a proteomic analysis of S2 and Kc167 cell EVs (*Koppen et al., 2011*). We think it likely that some or all these proteins are abundant contaminants that do not actually interact with Sas but happened to be present at higher levels in the IP from Sas-expressing cells *vs.* the IP from control cells. We did not further examine any of these proteins.

Since dArc1 was enriched in Sas$^{FL}$ preparations purified from EV lysates with anti-V5, we then investigated whether it binds to the Sas ICD (which would be in the EV interior) by co-IP in S2 cells. We coexpressed Myc epitope-tagged dArc1 with a fusion protein in which the V5-tagged ECD of mouse CD8 was attached to the TM domain and the 37 aa ICD of Sas. We then IP'd cell lysates with anti-Myc, and detected V5-mCD8$^{ECD}$-Sas$^{TM-ICD}$ and dArc1-Myc by western blotting. We observed that dArc1 co-IP'd with the Sas ICD fusion protein (*Figure 4c*).

The Sas ICD sequence contains the sequence motif YDNPSY, which is a PTB-binding motif (NPXY) that overlaps by two amino acids with an SH2-binding motif (YXXP) that is also a potential Abl tyrosine

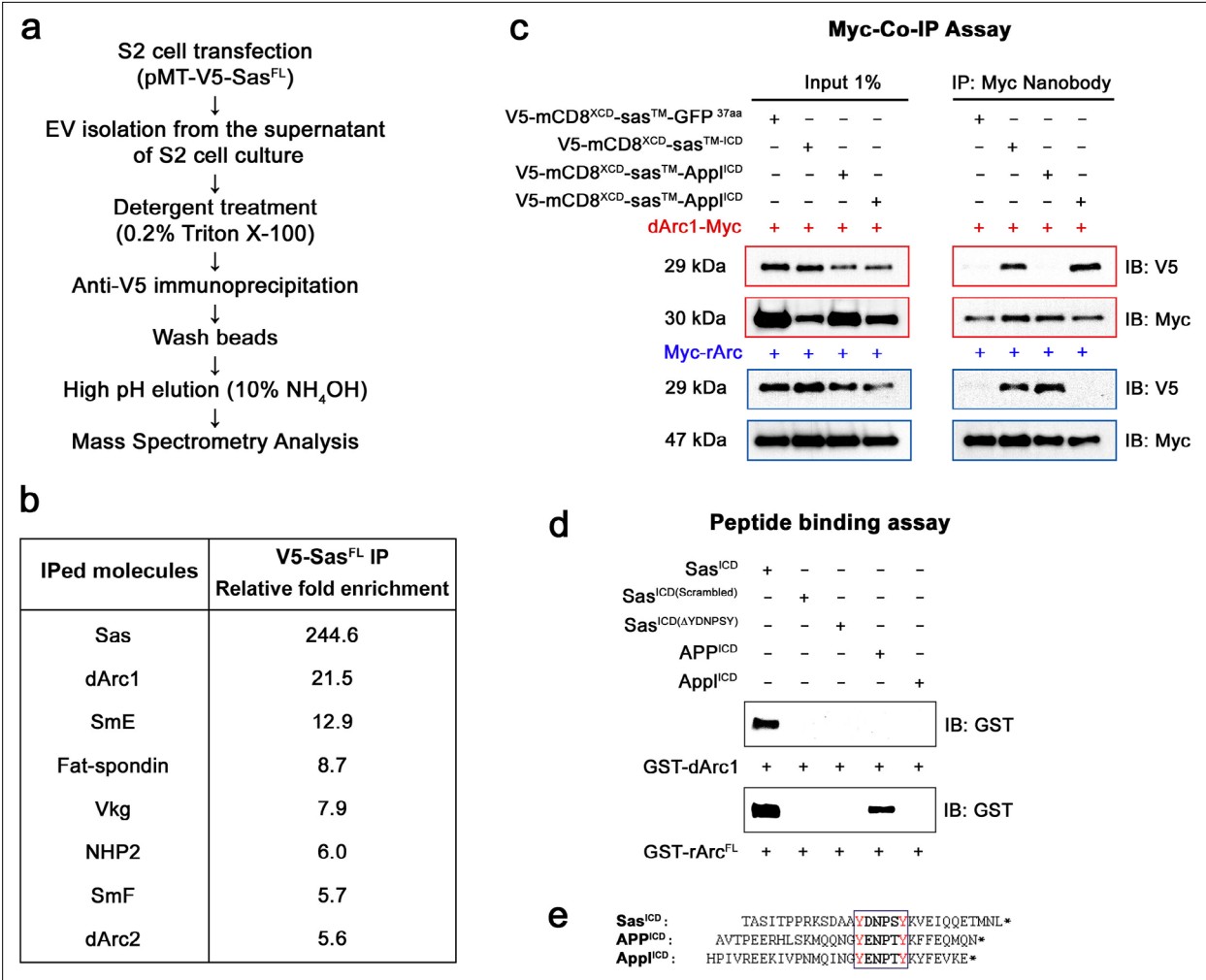

**Figure 4.** Interactions of Sas, Appl, and APP with Arcs. (**a**) Protocol for mass spectrometry analysis. Purified EVs from control S2 cells or S2 cells expressing V5-Sas^FL were lysed and IP'd with anti-V5, followed by protease digestion and mass spectrometry analysis. (**b**) Mass spectrometry results. The 7 proteins present at the highest levels in IPs from V5-Sas^FL EVs relative to IPs from control EVs (>6 fold ratio) are listed. Sas itself was the most highly enriched protein, as expected. dArc1 and dArc2 were enriched by 22-fold and 6-fold, respectively. (**c**) Co-IP/western blot analysis of association between Sas and Arc fusion proteins in transfected S2 cells. S2 cells were transfected with the V5-mCD8^ECD-Sas^TM-ICD fusion protein construct, or with equivalent constructs in which the Sas ICD was replaced by the Appl or APP ICD, with or without Myc-tagged dArc1 or mammalian (rat) Arc (rArc^FL) constructs. Lysates (Input 1%) were blotted with anti-Myc and anti-V5 (left), and IP'd with anti-Myc nanobody and blotted with anti-V5 and anti-Myc (right). Anti-V5 bands of the correct size were observed in anti-Myc IPs when dArc1-Myc was expressed with Sas or Appl ICD constructs (red symbols and boxes), and when Myc-rArc^FL was expressed together with Sas or APP ICD constructs (blue symbols and boxes)(n=6). (**d**) Direct binding of purified GST-dArc1 and GST-rArc^FL fusion proteins to Sas, APP, and Appl ICD peptides, as well as to scrambled and YDNPSY deletion mutant Sas peptides. Biotinylated peptides were bound to streptavidin magnetic beads, which were incubated with GST-Arc proteins, followed by western blotting of bead-bound proteins with anti-GST. GST-dArc1 bound to the wild-type, but not to scrambled or YDNPSY deletion mutant Sas ICD peptides, while GST-rArc^FL bound to wild-type Sas and APP ICD peptides. (**e**) Sequences of the complete Sas, APP, and Appl ICDs, corresponding to biotinylated peptide sequences. The conserved tyrosine motif is boxed, with tyrosines in red. *, stop codons.

The online version of this article includes the following source data for figure 4:

**Source data 1.** Source data files include raw and labelled images for the western blots shown in panels (c) and (d) and an Excel file of the table in panel (b).

**Source data 2.** Co-IP analyses raw data.

**Source data 3.** Peptide binding assay raw data.

**Source data 4.** MS analysis result table.

kinase substrate sequence (*Colicelli, 2010*; *Figure 4e*). The NPXY motif is the target for binding of the Numb PTB (*Li et al., 1998*). This suggests that an SH2 protein and a PTB protein might compete for binding to this sequence, if the first tyrosine was phosphorylated to create an SH2 docking site. The PTB domain of Numb does not require tyrosine phosphorylation to bind to its NPXY target. Interestingly, in an earlier mass spectrometric analysis, we found that the Shc protein, which contains a phosphotyrosine-binding SH2 domain, was associated with Sas purified from S2 cells treated with pervanadate to induce high-level tyrosine phosphorylation.

We searched for other *Drosophila* CSPs containing a sequence with similar properties in their ICDs, and found only one, Appl, which has the sequence YENPTY but is otherwise unrelated to the Sas ICD. Human APP, the mammalian ortholog of Appl, contains the same sequence in its short ICD (*Figure 4e*), as do the two APP paralogs, APLP1 and APLP2. We then replaced the Sas ICD in the V5-mCD8$^{ECD}$-Sas$^{TM-ICD}$ construct with the Appl and APP ICDs, and found that the Appl ICD protein co-IP'd with dArc1-Myc (*Figure 4c*), implicating the Y(D/E)NP(S/T)Y sequence in binding to dArc1. This sequence contains the consensus motif for binding of mammalian Arc to TARPγ2, CaMKII, and NMDA receptor peptides, which is X-P-X-(Y/F/H)(*Zhang et al., 2015*; *Nielsen et al., 2019*). Arc binds to the NMDA receptor as a monomer (*Nielsen et al., 2019*). The TARPγ2 Arc-binding peptide is RIPSYR, which is similar to the sequences in Sas (PSYK) and APP (PTYK). Accordingly, we expressed Myc-tagged mammalian Arc (rArc$^{FL}$) in S2 cells and examined whether it could co-IP with the V5-mCD8-ICD fusion proteins. We observed that Myc-rArc$^{FL}$ was able to co-IP with the Sas and APP ICDs (*Figure 4c*).

The co-IP data indicate that the Sas, APP, and Appl ICDs associate with dArc1 and Arc, but does not show that the proteins bind directly to each other. To evaluate this, we made the complete Sas, APP, and Appl ICDs (*Figure 4e*), as well as a scrambled version of the Sas ICD and a deletion mutant of the Sas ICD that lacks the YDNPSY sequence, as biotinylated peptides, and bound these to streptavidin-coupled magnetic beads. To make purified Arc proteins for binding, we expressed dArc1 and mammalian Arc as GST fusion proteins in *E. coli*. We mixed the beads with purified GST-dArc1 and GST-rArc$^{FL}$ proteins and examined whether we could observe specific binding. As a positive control, we made purified Numb PTB domain, and showed that it bound as expected to the Sas, APP, and Appl peptides, which all contain the NPXY PTB-binding motif, but not to the scrambled Sas peptide or the YDNPSY deletion mutant. In the peptide binding assay, we observed that dArc1 directly bound to the Sas ICD peptide, but not to the other peptides. Mammalian Arc also bound to the Sas ICD peptide, as well as to the APP ICD peptide (*Figure 4d*).

Mammalian and *Drosophila* Arc are not orthologs, and are apparently derived from independent Ty3/gypsy retrotransposon lineages (*Ashley et al., 2018*; *Pastuzyn et al., 2018*; *Hantak et al., 2021*). The fact that both proteins mediate intercellular communication suggests that they may be products of convergent evolution. Our results implicate the Y(D/E)NP(S/T)Y sequence as a determinant of binding to Arcs (*Figure 4e*), and show that fly and mammalian Arcs can bind to the same peptide sequences.

Our data also suggest that APP might be a CSP that has a relationship to Arc which is similar to that of Sas to dArc1. APP also localizes to EVs (*Laulagnier et al., 2018*; *Pérez-González et al., 2020*). The connection between APP and Arc will be of interest to explore in future studies, since Arc has been implicated in AD pathogenesis (*Wu et al., 2011*; *Landgren et al., 2012*; *Bi et al., 2018*). The first Y in the YENPTY motif in APP has been reported to be a substrate for the Abl tyrosine kinase (*Zambrano et al., 2001*). If YENP was phosphorylated, it would become a docking site for a class of SH2 domain proteins, and binding of this protein(s) could occlude Arc binding to the adjacent PTYK sequence. The Abl inhibitor imatinib (Gleevec), which would be expected to block phosphorylation of this site, inhibits formation of β-amyloid peptide (Aβ)(*Netzer et al., 2003*), and binding of Arc to APP could be relevant to this effect.

Excel file of mass spectrometry data. Contains two tabs: (1) Raw MaxQuant File; (2) Peptide and LFQ values.

## Sas can facilitate intercellular transfer of dArc1 and its mRNA in vivo

Sas is not required for loading of dArc1 capsids into EVs, since *dArc1* mRNA is a normal component of EVs from cell lines that do not express Sas. If it behaves like mammalian Arc in its interactions with peptides (*Nielsen et al., 2019*), dArc1 might bind to Sas as a monomer. Perhaps Sas recruits dArc1 monomers (possibly bound to mRNA *via* their Zn$^{2+}$ fingers) to nascent EVs during their biogenesis,

and they then assemble into capsids. Binding of Sas to dArc1 may help to increase the probability that Sas-bearing EVs contain dArc1 capsids bearing *dArc1* mRNA. One function of Sas might then be to deliver the EVs and their dArc1 capsid cargo, including encapsulated RNA, to specific recipient cells that express the Sas binding partner Ptp10D.

Having shown that Sas$^{FL}$ can move within larvae and that it binds to dArc1, which is a known component of EVs that mediates intercellular communication, we then examined whether it can cause dArc1 to move from source cells into recipient cells in vivo. To establish an assay system for dArc1 capsid movement in embryos, we first expressed V5-Sas$^{FL}$ in late stage 16 embryonic SGs together with RFP, and observed that V5 signal moved to the gut and tracheae, while RFP was retained in the SGs as expected. When V5-Sas$^{short}$ was expressed in SGs, however, V5 staining was only observed in the SGs (*Figure 5—figure supplement 1*). This is consistent with the failure of V5-Sas$^{short}$ to move from SGs to imaginal discs in larvae (*Figure 3c*).

To examine dArc1 protein transport, we needed to express untagged dArc1 and visualize it with antibody against dArc1 (*Ashley et al., 2018*), because we were unsuccessful in detecting movement of tagged versions of dArc1. dArc1 is made at very low levels in embryos. In late stage 16 control embryos (*Sage-GAL4/+* or *Sage-GAL4 >Sas$^{FL}$*), we observed faint staining throughout the embryo, with higher levels in the gut (*Figure 5—figure supplement 2*). We then expressed dArc1 from a UAS construct that contained only the dArc1 open reading frame (ORF), flanked by heterologous 5' and 3' UTR sequences. The 3' UTR was derived from SV40. When we expressed dArc1 alone in SGs, we observed bright anti-dArc1 staining in the SGs and increased staining relative to controls in the gut and in dots in the body wall. When Sas$^{FL}$ and dArc1 were expressed together, dArc1 staining in the gut was further increased (*Figure 5—figure supplement 2*).

To localize dArc1 staining in the body wall and compare it to Ptp10D staining, we examined dissected 'fillets' at high magnification. For reference, *Figure 5—figure supplement 1* shows the evolution of Ptp10D expression in fillets from stage 14 to late stage 16. VNC expression continuously increases during this time period, while tracheal expression begins in stage 14, decreases in stage 15, and re-emerges at stage 16, at which time Ptp10D is expressed in the main tracheal trunk and major tracheal branches. *Figure 5d'* shows that, in late stage 16 embryos expressing both Sas$^{FL}$ and dArc1 in SGs, there were many bright puncta stained with anti-dArc1 in the dorsal tracheal trunk, which expresses Ptp10D. These puncta appeared similar to those previously observed at larval NMJs (*Ashley et al., 2018*). They were not detectable in control embryos (*Sage-GAL4/+; Figure 5a'*).

There were lower numbers of fainter dArc1 puncta in tracheal trunks of the two other genotypes (*Sage >dArc1* and *Sage >Sas$^{FL}$*)(*Figure 5b' and c'*). Endogenous Sas is expressed at low levels in SGs, and endogenous *dArc1* mRNA is also present in SGs (*Figure 5g*), although dArc1 protein is not detectable. Endogenous Sas$^{FL}$ may be able to transport some of the overexpressed dArc1, and overexpressed Sas$^{FL}$ might transport some endogenous dArc1, giving rise to the observed puncta. It is also interesting that dArc1 (and *dArc1* mRNA; see below) is observed in tracheal cells, but not in VNC neurons, which also express Ptp10D at high levels. There is a glial sheath around the VNC at late stage 16, and this might block access of EVs to Ptp10D-expressing neurons. Alternatively, perhaps there are cofactors required for EV binding and/or internalization that are not expressed in neurons.

Dramatic effects of Sas$^{FL}$ on *dArc1* expression and localization were observed when endogenous *dArc1* mRNA was examined by fluorescence in situ hybridization (FISH) in embryos expressing the UAS-dArc1 ORF construct in SGs. To detect mRNA, we used the 700 nt antisense 3' UTR probe employed in the (*Ashley et al., 2018*) paper to visualize *dArc1* mRNA puncta at the NMJ. Note that this probe does not recognize overexpressed *dArc1* mRNA made from the UAS construct, because that contains only the dArc1 ORF and no *dArc1* 3' UTR sequences. In late stage 16 control embryos (*Sage-GAL4/+*), we observed faint FISH signals in the SGs and a few puncta elsewhere in the embryo (*Figure 5e*). A similar pattern was seen in *Sage >Sas$^{FL}$* embryos (*Figure 5g*). When dArc1 was expressed from the UAS-dArc1 ORF construct, we observed bright FISH signals in SGs with the endogenous *dArc1* 3' UTR probe (*Figure 5f*). There were also scattered puncta in other parts of the embryos. This shows that exogenous dArc1 induces expression of endogenous *dArc1* mRNA (or stabilizes the mRNA). No signal was observed when a sense *dArc1* 3' UTR probe was used for FISH (*Figure 5—figure supplement 2*). Finally, when Sas$^{FL}$ and dArc1 were expressed together, we observed a completely different pattern, in which the entire tracheal system is lit up by the FISH signal for the endogenous *dArc1* 3' UTR (*Figure 5h*). The foregut and esophagus also stain brightly. By contrast, in embryos expressing Sas$^{short}$

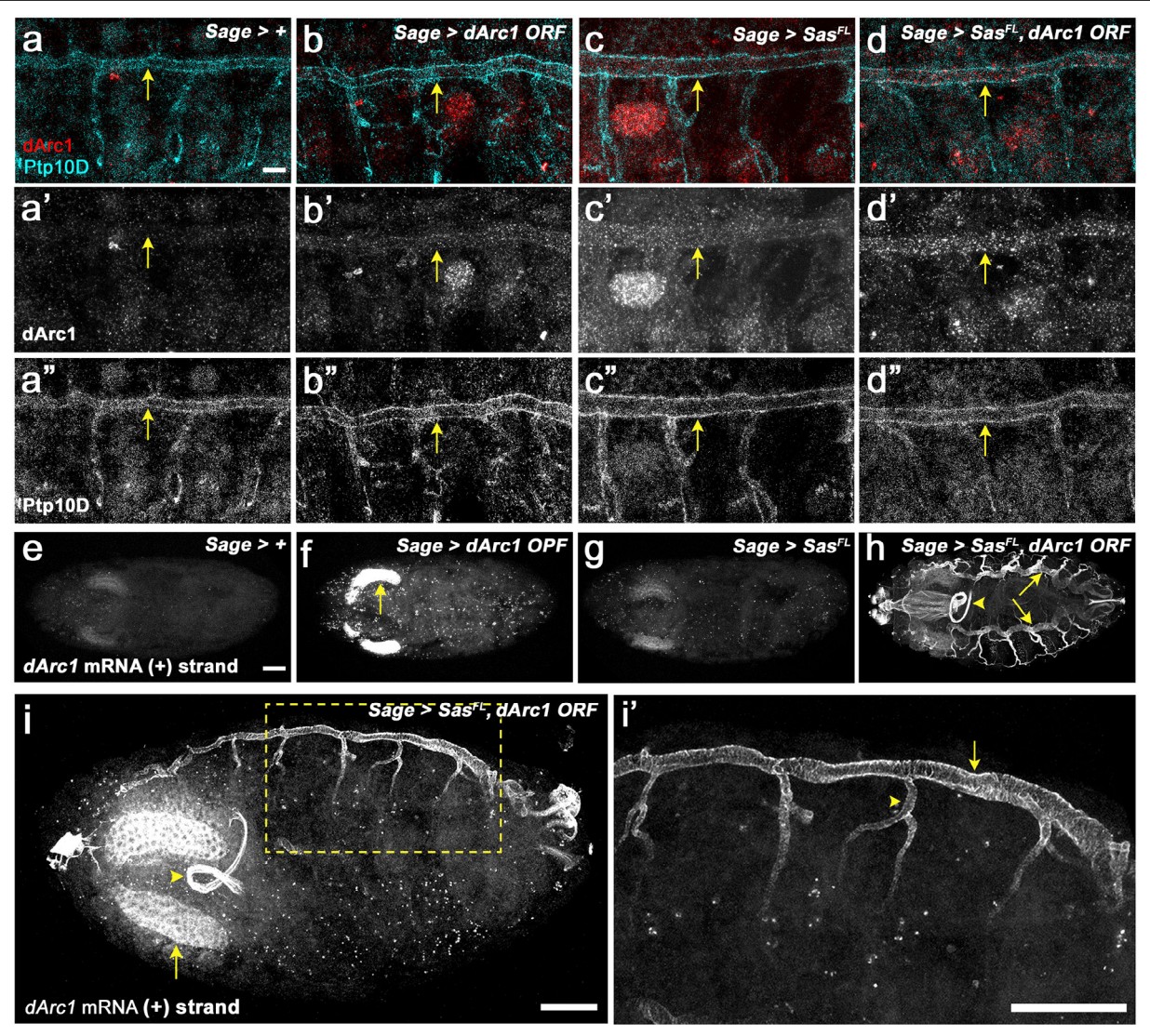

**Figure 5.** Sas facilitates transfer of dArc1 capsids bearing dArc1 mRNA into distant cells in vivo. (**a-d**) Localization of dArc1 protein and Ptp10D in high-magnification views of body walls from fillets of late stage 16 embryos (anterior to the left, dorsal up). (**a**) Control (*Sage-GAL4/+*); (**b**) *Sage >dArc1*; (**c**), *Sage >Sas^FL^*; (**d**), *Sage >Sas^FL^ + dArc1*. (**a–d**) show double-staining with anti-dArc1 (red) and anti-Ptp10D (blue). (**a'-d'**) show the dArc1 channel alone. (**a"-d"**) show the Ptp10D channel alone. Arrows, dorsal tracheal trunk. There are numerous bright dArc1 puncta in the tracheal trunk when Sas^FL^ and dArc1 are expressed together. Fewer and weaker puncta are observed when Sas^FL^ or dArc1 are expressed alone, and no puncta are seen in *Sage-GAL4/+* controls. (**e–i**) *dArc1* mRNA from the endogenous gene (+strand), detected by FISH with a 3' UTR minus-strand (antisense) probe. (**e**) Control (*Sage-GAL4/+*); (**f**) *Sage >dArc1*; (**g**) *Sage >Sas^FL^*; (**h**) *Sage >Sas^FL^ + dArc1*. There is weak expression of *dArc1* mRNA in the SGs in controls. When dArc1 (from an ORF construct) is expressed alone, bright SG staining is observed, indicating that exogenous dArc1 increases expression of endogenous *dArc1* mRNA. There are also scattered *dArc1* mRNA puncta elsewhere in the embryo. When Sas^FL^ and dArc1 are expressed together, bright *dArc1* mRNA FISH staining of the entire tracheal system is observed (arrows indicate dorsal tracheal trunks), as well as the foregut (arrowhead) and esophagus (n>100 embryos examined for each genotype; representative results are shown). **i, i'**, high-magnification views of *dArc1* mRNA in the tracheae in an obliquely mounted (anterior to the left, dorsal up) embryo expressing Sas^FL^ and dArc1 in SGs. **i'** is a higher-magnification inset (yellow dotted outline) from (**i**). Arrow in (**i**), SG; arrowhead, foregut loop. Arrow in **i'**, dorsal tracheal trunk; arrowhead, transverse connective. Scale bar in (**e**) (applies to **e–h**), 50 μm; scale bar in (**a**) (applies to **a–d**), 10 μm; scale bar in (**i**), 50 μm; scale bar in (**i'**), 50 μm.

The online version of this article includes the following figure supplement(s) for figure 5:

**Figure supplement 1.** Ptp10D expression in *wt*, and V5-Sas^FL^ and V5-Sas^short^ localization in embryos in which they are expressed from *Sage-GAL4*.

**Figure supplement 2.** dArc1 protein localization, and additional FISH results with the SV40 3' UTR and *dArc1* 3' UTR antisense and sense (control) probes.

and dArc1 in SGs, the *dArc1* FISH signal is observed in the SGs, with only a few puncta elsewhere in the embryo (*Figure 5—figure supplement 2*), consistent with the fact that Sas^short cannot move within the embryo (*Figure 5—figure supplement 1*).

When an antisense probe for the SV40 3' UTR sequence in the UAS-dArc1 construct was used for FISH, we observed signal only within the SGs, even when Sas^FL was coexpressed (*Figure 5—figure supplement 2*). This suggests that *dArc1* mRNA lacking the endogenous *dArc1* 3' UTR sequences is not efficiently loaded into dArc1 capsids that can move elsewhere in the embryo. This is consistent with the findings of *Ashley et al., 2018*, who showed that the *dArc1* 3' UTR is required for *dArc1* mRNA transfer at the NMJ. No signal was observed when a sense SV40 3' UTR probe was used for FISH (*Figure 5—figure supplement 2*).

*Figure 5i and i'* show the tracheae and SGs at higher magnification in side views of an embryo expressing both Sas^FL and dArc1 in SGs. The dorsal tracheal trunk (arrow) and the transverse connective (arrowhead) both display bright *dArc1* FISH signals. Note that, because this is a confocal image (optical section), the cells at the edges of the tracheal trunk are bright, while the hollow lumen is dark. The brightness of the tracheal FISH signal suggests that it represents not only *dArc1* mRNA transferred from capsids, but *dArc1* mRNA synthesized in these cells in response to dArc1 protein made from the *dArc1* mRNA transported in the capsid. If this is correct, it would represent an amplification

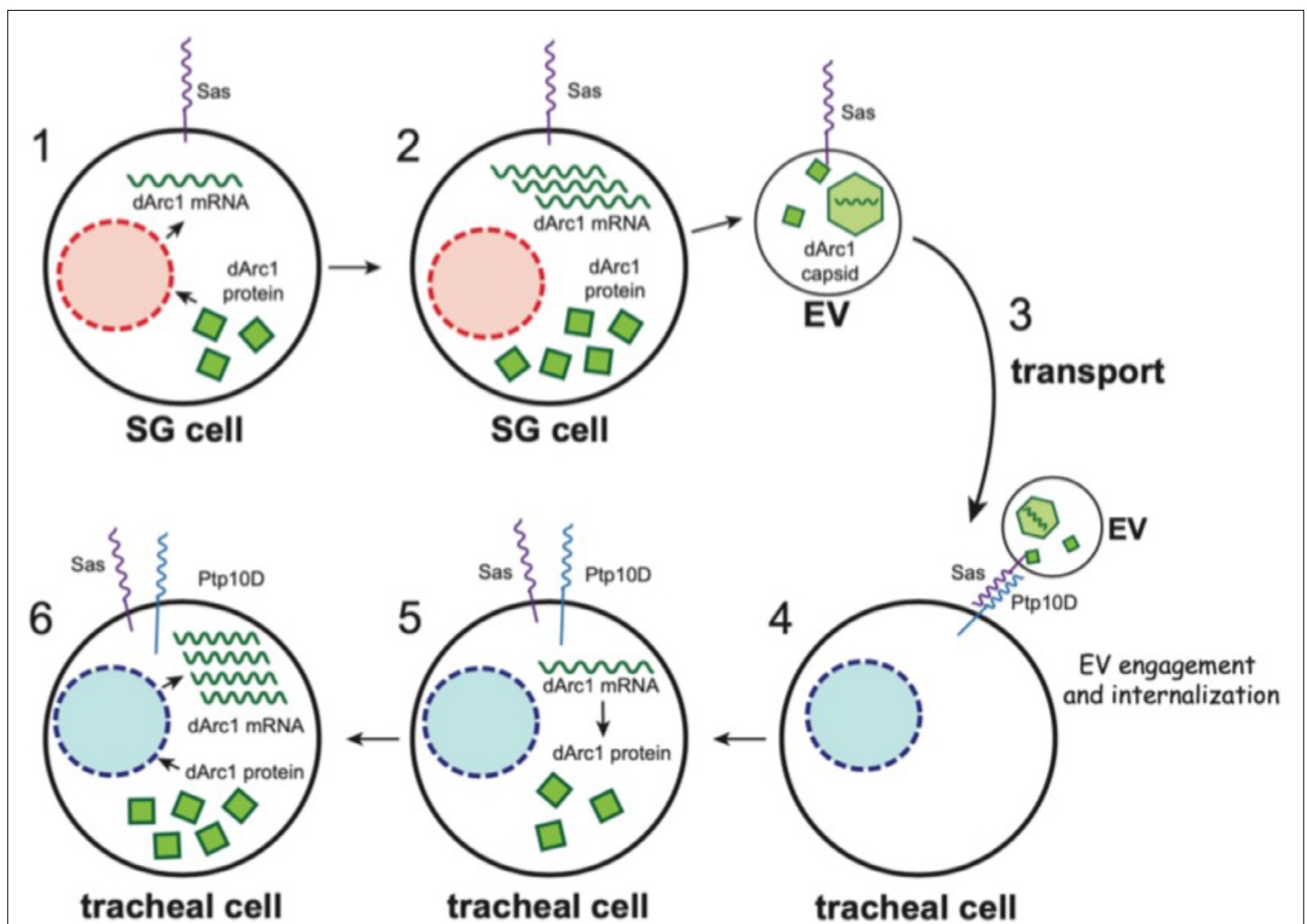

**Figure 6.** Schematic diagram of the processes involved in movement of EVs bearing Sas^FL and dArc1 capsids from salivary glands to tracheal cells. Steps 1 and 2, the presence of the dArc1 protein made from a UAS-dArc1 ORF construct increases expression of endogenous *dArc1* mRNA in embryonic SGs. EVs with Sas^FL on their surfaces bearing dArc1 capsids containing endogenous *dArc1* mRNA diffuse or are transported through the hemolymph (Step 3) and bind to Ptp10D-expressing tracheal cells (Step 4). The EVs internalize into the tracheal cells and release *dArc1* mRNA (Step 5), and dArc1 protein translated from that mRNA induces expression of more endogenous *dArc1* mRNA (Steps 5 and 6). Nuclei, orange circles in 1–2 and blue circles in 4–6.

mechanism in which translated *dArc1* mRNA from EVs can induce expression of much more *dArc1* mRNA in the recipient cells. Finally, we examined whether the Sas ICD is required for *dArc1* mRNA transport by expressing dArc1 together with a protein (Sas^ECD-TM-GFP) in which the Sas ICD was replaced by GFP. This protein is present in EVs when expressed in S2 cells, but it does not produce any *dArc1* FISH signal outside of the SGs (*Figure 5—figure supplement 2*), indicating that it cannot facilitate transport of dArc1 capsids to tracheal cells.

## Conclusions

Our results on movement of Sas EVs containing dArc1 capsids are summarized in the diagram of *Figure 6*. These findings contribute to the understanding of intercellular communication mechanisms by showing that Sas^FL is an EV targeting ligand that can direct internalization of EVs into cells expressing the Sas receptor Ptp10D. Endogenous Sas^FL is likely to move between cells in vivo, because tagged Sas^FL moves away from both neuronal and non-neuronal cells when ectopically expressed.

dArc1 is related to retrotransposon Gag proteins, and it forms a capsid that contains *dArc1* mRNA and is loaded into EVs (*Ashley et al., 2018*). Overexpressed Sas^FL can facilitate transfer of dArc1 capsids into distant Ptp10D-expressing recipient cells in vivo. Whether endogenous Sas^FL also transfers dArc1 between cells cannot be determined from our data, and would be difficult to evaluate, since both the *sas* and *dArc1* genes are broadly expressed. The Sas ICD binds directly to dArc1.

Mammalian Arc also forms capsids that are transported *via* EVs (*Pastuzyn et al., 2018*), and it binds to the Sas and APP ICDs, which share a tyrosine motif. Full-length APP and some of its proteolytic products are localized to EVs, and EVs from N2a cells bearing tagged APP are internalized into cultured neurons, but not into glia (*Laulagnier et al., 2018*). It will be interesting to determine if APP EVs contain Arc capsids, and if the presence of APP on Arc-containing EVs causes Arc to be preferentially delivered to a specific population of neurons. This could have implications for Alzheimer's disease research, because APP is the source of β-amyloid peptide and Arc has been linked to β-amyloid accumulation and AD pathogenesis (*Wu et al., 2011*; *Landgren et al., 2012*; *Bi et al., 2018*).

## Methods
### Fly stocks and genetics

The following stocks were used: *yw* for wild-type control, *ap-GAL4* (Bloomington 50156), *UAS-mCD8::GFP* (Bloomington 5130), *UAS-myr::mRFP* (Bloomington 7118), *UAS-mCherry.NLS* (Bloomington 38424), *sas15* (null mutant)(Bloomington 2098), *Sage-GAL4* (a gift from Deborah J. Andrew), *Ptp10DEP1172* (Bloomington 11332), *UAS-dArc1* (Bloomington 37532), *UAS-Numb* (a gift from Yuh Nung Jan), *UAS-SasFL* and *UAS-V5-SasFL* (*Lee et al., 2013*), *Arc1esm18* (Bloomington 37530). Crosses and embryo collections were performed at room temperature. For overexpression experiments, embryos were shifted to 29 °C for at least 120 min prior to fixation and staining and 3rd instar larvae were shifted to 29 °C for overnight for further analysis. For the EV targeting experiments, imaginal discs from 3rd instar larvae were harvested at room temperature and incubated in 200 µl of S2 supernatant overnight at 29 °C before fixation and staining. There are 10,000–50,000 cells in a 3rd instar imaginal disc. Given the results from the NTA analysis, we can conclude that ~140,000 EVs are present in 200 µl of supernatant from V5-Sas^FL-expressing S2 cells. We used 5 wing discs per incubation, so the ratio of EVs to cells is ~0.5 to~2. The relative V5 signal intensities on the imaginal discs were measured by densitometry analysis using ImageJ software.

### Immunohistochemistry

Embryos and larval tissues were stained with standard immunohistochemical procedures. The following antibodies were used: rabbit anti-V5 (1:1000, Invitrogen); mouse anti-GFP (1:1000, Invitrogen); rabbit-anti-Sas^FL (1:2000, gift of D. Cavener); rat-anti-Sas^short (1:50, GenScript USA Inc); mAb Cq4 against crumbs (1:100, DSHB); guinea pig-anti-Numb (1:1000, gift from J. Skeath); rabbit-anti-dArc1 (1:100, gift from T. Thomson); mAb 8B2 against Ptp10D (1:5, DSHB); mAb MR1A against Prospero (1:40, DSHB); rat-anti-Repo (1/2000, gift from S. Banerjee); rabbit anti-Evi (Wntless, 1:5000, gift from K. Basler); FITC-conjugated phalloidin (1:1000, Thermo Fisher Scientific); AlexaFluor 488 anti-mouse, AlexaFluor 488 anti-rat, AlexaFluor 568 anti-rabbit, AlexaFluor 568 anti-rat and AlexaFluor 647 anti-mouse (1:1000, Invitrogen). Rat anti-Sas^short antibody was generated against a synthetic peptide,

HSSIPANGANNLQP, flanking the EVT region (intron is between the N and G residues) and the KLH-conjugated antibody was purified by protein G column (GenScript USA Inc). Samples were mounted in VECTASHIELD (Vector Laboratories) and analyzed on a Zeiss LSM 880.

## Cell culture and preparation of EVs and cell lysates

EVs and cell lysates were prepared from S2 cells (ATCC, CRL-1963) that were cultured for four days at 22°C in Schneider's medium (Gibco) supplemented with 10% exosome-free FBS (#EXO-FBSHI-50A-1, SBI) to avoid contamination from Bovine serum exosomes. S2 cells were authenticated and confirmed to be free of mycoplasma by ATCC. DNA constructs were transiently transfected into S2 cells using Effectene (Qiagen). EVs for western blot analysis and electron microscopy were collected using Total Exosome Isolation reagent (#4478359, Invitrogen) from the supernatants of S2 cultures. This kit has been found to produce exosomes of equivalent quality from mammalian cells (with respect to the presence of exosome markers and the depletion of non-exosome proteins) to those generated using ultracentrifugation (*Skottvoll et al., 2019*). One part of the reagent and two parts of supernatant were mixed and incubated at 4°C overnight. Pellets of EVs were collected after centrifugation at 10,000 x *g* for 60 min at 4°C. The EV pellets were resuspended in PBS for western blot analysis.

To prepare EVs using the ultracentrifugation protocol of *Théry et al., 2006*, S2 cells were cultured in a medium with 10% exosome-free FBS (SBI) and the cultures were grown in 100 mm Petri dishes at 22°C. Cells were collected at 1–1.5×10⁶ cells/ml. The supernatant from a 50 ml culture was filtered using a 0.22 µm filter to eliminate dead cells and large debris prior to further purification by ultracentrifugation (Optima MAX-XP, Beckman Coulter). The filtered supernatants were spun at 100,000 x *g* for 70 min and the collected EV pellets were resuspended in PBS. Then, the samples were spun again at 100,000 x *g* for 70 min. The final EV pellet was resuspended in 60 µL of PBS and processed for further analyses or stored at –80°C.

For the EV targeting experiments between S2 cells, supernatants from transiently transfected donor cells were collected and filtered using 0.22 µm PVDF membranes before resuspension and incubation with the recipient cells. Two days before the supernatant swap between EV donor and recipient cell cultures, the recipient cells were transiently transfected with DNA constructs. The recipient cells were incubated in the supernatants with EVs from donor cells for 2 hr at 22°C. For western blot analyses, cell lysates were prepared using RIPA cell lysis buffer. To measure the size and number of EV particles from S2 cell culture, collected EV pellets were subjected to NTA by System Biosciences, LLC (Palo Alto, CA, USA) using a NanoSight instrument. The NTA measurements rely on light scattering to extract particle size and the number of particles in a sample and the NTA software (Version 2.3) collects data on multiple particles to calculate the hydrodynamic diameter of each particle using the Stokes-Einstein equation (System Biosciences, LLC).

## Mass spectrometry analysis

Samples were lyophilized and proteins were trypsin-digested as previously described (*Pierce et al., 2013*). A total of 200 ng of digested peptides were analyzed as previously described (*Sung et al., 2016*). Briefly, peptides were loaded onto a 26 cm analytical HPLC column (75 µm inner diameter) packed with ReproSil-Pur C$_{18AQ}$ 1.9 µm resin (120 Å pore size; Dr. Maisch, Ammerbuch, Germany). Peptides were separated with a 120 min gradient at a flow rate of 350 nl/min at 50 °C (column heater) using the following gradient: 2–6% solvent B (7.5 min), 6–25% B (82.5 min), 25–40% B (30 min), 40–100% B (1 min), and 100% B (9 min), where solvent A was 97.8% H$_2$O, 2% ACN, and 0.2% formic acid, and solvent B was 19.8% H$_2$O, 80% ACN, and 0.2% formic acid. Samples were analyzed using an EASY-nLC 1000 coupled to an Orbitrap Fusion operated in data-dependent acquisition mode to automatically switch between a full scan ($m/z$=350–1500) in the Orbitrap at 120,000 resolving power and an MS/MS scan of higher-energy collisional dissociation fragmentation detected in the ion trap (using TopSpeed). The automatic gain control (AGC) targets of the Orbitrap and ion trap were 400,000 and 10,000.

## Mass spectrometry data

Raw data were searched using MaxQuant (version 1.5.3.30)(*Cox and Mann, 2008*; *Wagner et al., 2011*) against the Uniprot D *melanogaster* database. Fragment ion tolerance was 0.5 Da. Precursor mass tolerance was 4.5 ppm after automatic recalibration. Searches were permitted up to two missed

tryptic peptide cleavages. Cysteine carbamidomethylation was designated as a fixed modification while Methionine oxidation and N-terminal acetylation were designated as variable modifications. False discovery rates were estimated to be <1% using a target-decoy approach. Complete data are in *Supplementary file 1*.

## Protein expression and purification

To express and purify Arc proteins in the *E. coli* system, the cDNAs of dArc1, dArc2 and rArc were subcloned into the pGEX-4T-1 vectors together with GST-6xHis-tags and TEV protease cleavage site. Arc proteins were expressed in *E. coli* strain BL21 (DE3) grown in LB broth by induction of log-phase cultures with 1 mM isopropyl-β-D-thiogalactopyranoside (IPTG) and incubated overnight at 23 °C. Cells were pelleted and resuspended in B-PER lysis buffer (#78243, Thermo Fisher Scientific) before centrifugation to collect cell lysates.

Tagged Arc proteins were pulled down using Ni-NTA resin column and the eluates with GST-6xHis-dArc1 and GST-6xHis–rArc proteins were used for peptide binding assays.

## Western blotting

Proteins were separated by SDS–PAGE, transferred at 200 mA for 60 min to nitrocellulose membranes using a Bio-Rad Wet Tank Blotting System in Tris-Glycine Transfer Buffer with 10% methanol. Blocked membranes were incubated with primary antibodies in 0.5% milk PBS-0.1% Tween for overnight. HRP-conjugated antibodies (anti-V5-HRP (#RV5-45P-Z, ICL), anti-mouse IgG HRP (#sc-516102, Santa Cruz Biotechnology), anti-beta-actin-HRP (#HRP-60008, Proteintech), anti-rabbit IgG HRP (#65–6120, Invitrogen), anti-rat IgG HRP (#35470, Invitrogen), anti-alpha-tubulin-HRP (#HRP-66031, Proteintech), anti-cMyc-HRP (#RMYC-45P-Z, ICL), and anti-GST-HRP (#MA4-004-HRP, Invitrogen)) were used at 1:10,000 for 60 min. Blots were developed using ECL Western Blotting Substrate (#32109, Pierce), and imaged on a MINI-MED 90 X-Ray Film Processor (AFP Manufacturing Co.).

## Electron tomography and immuno-EM

For imaging of EVs by electron tomography (ET), EVs were prepared using the Exosome Isolation Kit as described above. Supernatant was removed and replaced with ~10 ml 10% Ficoll, 5% sucrose in 0.1 M sodium cacodylate trihydrate with minimal disturbance of the pellet. Pellets were transferred to brass planchettes (type A/B; Ted Pella, Inc) and ultra-rapidly frozen with a HPM-010 high-pressure freezing machine (Bal-Tec/ABRA). Vitrified samples were transferred under liquid nitrogen to cryo-tubes (nunc) containing a frozen solution of 2.5% osmium tetroxide, 0.05% uranyl acetate in acetone and placed in an AFS-2 Freeze-Substitution Machine (Leica Microsystems, Vienna). Samples were freeze-substituted at –90 °C for 72 hr, warmed to –20 °C over 12 hr, held at –20° for 12 hr, then warmed to room temperature. Samples were rinsed 3 x with acetone and infiltrated into Epon-Araldite resin (Electron Microscopy Sciences). Resin was polymerized at 60 °C for 24 hr.

Serial semi-thin (170 nm) sections were cut with a UC6 ultramicrotome (Leica Microsystems) using a diamond knife (Diatome Ltd., Switzerland). Sections were collected onto Formar-coated copper/rhodium slot grids (Electron Microscopy Sciences) and stained with 3% uranyl acetate and lead citrate. Colloidal gold particles (10 nm) were placed on both surfaces of the grid to serve as fiducial markers for subsequent image alignment. Grids were placed in a dual-axis tomography holder (Model 2040; Fischione Instruments, Inc) and imaged with a Tecnai T12 transmission electron microscope (Thermo-Fisher Scientific) at 120 k eV. For dual-axis tomography, grids were tilted +/-62° and images acquired at 1° intervals. The grid was rotated 90° and a similar tilt-series was recorded about the orthogonal axis. Tilt-series data was acquired automatically using the SerialEM software package. Tomographic data was calculated, analyzed and modeled on iMac Pro and M1 computers (Apple, Inc) using the IMOD software package.

For immuno-EM, EV pellets were prepared as per above. Supernatant was removed and pellets fixed with 4% paraformaldehyde in PBS for 1 hr. Pellets were then infiltrated with 2.1 M sucrose in PBS over 24 hr, with >3 changes of the infiltration solution during that time. Pellets were placed onto aluminum sectioning stubs, drained of excess liquid and frozen in liquid nitrogen. Cryosections (100 nm) were cut at –140 °C with a UC6/FC6 cryoultramicrotome (Leica Microsystems) using cryo-diamond knives (Diatome Ltd). Cryosections were collected with a wire loop containing 2.3 M sucrose in PBS and transferred to Formvar-coated, carbon-coated, glow-discharged 100-mesh copper/

rhodium grids (Electron Microscopy Sciences) at room temperature. Nonspecific antibody binding sites were blocked by incubating the grids with 10% calf serum in PBS for 30'. Sections were then labeled with 1° antibodies (diluted in 5% calf serum/PBS) for 2 hr, rinsed 4 x with PBS, then labeled with 10 nm and/or 15 nm gold-conjugated 2° antibodies (diluted in 5% calf serum/PBS) for 2 hr. Grids were rinsed 4 x with PBS, 3 x with dH$_2$O then simultaneously negatively stained and stabilized with 1% uranyl acetate, 1% methylcellulose in dH$_2$O. Immuno-EM samples were imaged as per the tomography samples, above.

## Immunoprecipitation

For the Myc-co-IP assay, transiently transfected S2 cells using Effectene (Qiagen) were cultured in Schneider's medium at 22 °C for 4 days. Tagged expression constructs (V5-mCD8$^{ECD}$-sas$^{TM-ICD}$, V5-mCD8$^{ECD}$-APP$^{ICD}$, V5-mCD8$^{ECD}$-Appl$^{ICD}$, dArc1-Myc and Myc-rArc) were cloned in pAc5.1B vector according to standard cloning procedure. For IP analysis, cell lysates were prepared using IP Lysis buffer (#87787, Pierce) and the lysates were incubated in Myc-Trap agarose (#yta-20, Chromotek) following the manufacturer's protocol and the eluates were analyzed by standard western blot analysis.

## Peptide binding assay

For the peptide binding assay, biotinylated peptides (wt Sas$^{ICD}$, Sas$^{ICD}$ variations (scrambled and ΔYDNPSY), APP$^{ICD}$ and Appl$^{ICD}$) made by RS Synthesis, Inc, were incubated with Streptavidin magnetic beads (#88817, Pierce) for 45 min at 4 °C and the beads were extensively washed with TBST. Purified GST-6xHis-dArc1 and rArc proteins were added to the beads with bound biotinylated peptides and incubated at 4 °C overnight. Similar experiments were performed with Numb PTB domain protein purified from *E. coli*. The beads were carefully washed with TBST and eluates prepared for western blot analysis following the standard protocol described above.

## Fluorescent in situ hybridization (FISH)

The FISH protocol was a modification of protocols from *Kosman et al., 2004*. Fixed L16 whole embryos were prepared using standard protocols and rinsed with ethanol quickly four times. Then the embryos were permeabilized twice with a mixture of xylenes and ethanol (1:2, v/v) and washed three times with ethanol for 5 min each. To rehydrate the embryos, the embryos were washed with 100%, 50% and 0% methanol in PBT sequentially for 30 min each step. The rehydrated embryos were permeabilized again using proteinase K (20 μg/mL in PBT) for exactly 7 min and washed three times for 5 min each in PBT followed by a second fixation (5% paraformaldyhyde and 1% DMSO in PBT) for 25 min and washed three times in PBT for 5 min each. Then the embryos were prepared for pre-hybridization by incubation in 50% hybridization buffer (50% formamide, 5 x SSC, 100 μg/ml fragmented salmon testes DNA, 50 μg/ml heparin, 0.1% Tween-20) in PBT for 5 min. For pre-hybridization, embryos were incubated in hybridization buffer for more than 90 min at 55°C while changing the buffer every 30 min. The pre-hybridized embryos were incubated in DIG-tagged dArc1 mRNA probe for 18 hr at 55°C for annealing. The embryos were washed with hybridization buffer three times for 30 min each at 55°C, after which the buffer was replaced with replaced the buffer with PBT containing rhodamine-conjugated sheep anti-DIG antibody (#11207750910, SigmaAldrich) overnight at 4°C. Then the embryos were washed and mounted for confocal microscopy.

## Probe preparation

Probes for detection of endogenous *dArc1* mRNA were designed against a 760 nt region of the *dArc1* mRNA 3' UTR sequence, which was used for FISH in a previous study (*Ashley et al., 2018*). Probes for detetion of *dArc1* mRNA from the UAS-dArc1 ORF construct were designed against a 616 nt region of SV40 3' UTR sequence. To generate antisense and sense probes for *dArc1* mRNA, cDNA sequences from *dArc1* were PCR amplified and purified to use as positive and negative probe templates. The same procedure was used for antisense and sense probes for the SV40 3' UTR. The DNA templates were heated to 55°C for 2 min and then put back on ice. Transcription reactions were set up to label probes with digoxigenin (DIG, # 11277073910, Roche) and incubated at 37°C for 2 hr. Probes were precipitated and resuspended in hybridization buffer and stored at –20°C.

The following primers were used to generate endogenous *dArc1* mRNA probes:

dArc1 probe forward primer: GATTTTTCGTCTGATCCTGGTC

dArc1 probe reverse primer: CCGTTTCTGAGTTTAATGGTTG

These primers were used to generate SV40 3' UTR mRNA probes:

SV40 3' UTR probe forward primer: TCTAGAGGATCTTTGTGA
SV40 3' UTR probe reverse primer: TGCTATTGCTTTATTTGT

## Acknowledgements

Mass spectrometry work was performed at the Caltech Proteome Exploration Laboratory. Imaging was done at the Caltech Biological Imaging facility. EM work was done at the Caltech Cryo-EM facility. We thank Violana Nesterova for figure preparation. We thank the following colleagues for reagents and *Drosophila* lines: Jason Shepherd (University of Utah) for pGEX-dArc and rArc constructs; Travis Thomson and Vivian Budnik (University of Massachusetts) for rabbit anti-Arc1; Douglas Cavener (Penn State) for rabbit anti-Sas^FL; Deborah Andrew (Johns Hopkins) for *Sage-GAL4*; James Skeath (Washington University) for guinea pig anti-Numb; Swati Banerjee (UTHSC, San Antonio) for rat anti-Repo, and Yuh-Nung Jan (UCSF) for *UAS-Numb*. We thank Simon Erlendsson, Fernando Bazan, Paul Worley, and Tino Pleiner for discussions about Arc purification and Arc and Sas structures. This work was supported by NIH RO1 grants NS28182 and NS096509 to KZ, and by Howard Hughes Medical Institute support to R Deshaies, who was JMR's faculty supervisor when he was a postdoctoral fellow at Caltech.

## Additional information

### Competing interests

Justin M Reitsma: is affiliated with AbbVie. The author has no other competing interests to declare. The other authors declare that no competing interests exist.

### Funding

| Funder | Grant reference number | Author |
|---|---|---|
| National Institute of Neurological Disorders and Stroke | NS28182 | Kai Zinn |
| National Institute of Neurological Disorders and Stroke | NS096509 | Kai Zinn |

The funders had no role in study design, data collection and interpretation, or the decision to submit the work for publication.

### Author contributions

Peter H Lee, Conceptualization, Resources, Data curation, Formal analysis, Validation, Investigation, Visualization, Methodology, Writing - original draft, Writing - review and editing; Michael Anaya, Investigation, Methodology; Mark S Ladinsky, Investigation, Visualization, Methodology; Justin M Reitsma, Resources, Software, Investigation, Methodology; Kai Zinn, Conceptualization, Supervision, Funding acquisition, Writing - original draft, Project administration, Writing - review and editing

### Author ORCIDs

Peter H Lee ⓘ http://orcid.org/0000-0003-2411-6094
Mark S Ladinsky ⓘ http://orcid.org/0000-0002-1036-3513
Kai Zinn ⓘ http://orcid.org/0000-0002-6706-5605

### Decision letter and Author response

Decision letter https://doi.org/10.7554/eLife.82874.sa1
Author response https://doi.org/10.7554/eLife.82874.sa2

## Additional files

### Supplementary files
• Supplementary file 1. Sas IP MS analysis raw data.

• MDAR checklist

### Data availability
Key data generated or analysed during this study are included in the manuscript and supporting files.

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
