## [Editor Report]

The manuscript addresses how extracellular vesicles (EV) are targeted to their recipient cells once they are produced and released. The study shows that a transmembrane protein Sas gets incorporated into EVs, and this protein binds to its receptor Ptp10D on target cells, thus targeting the EVs. The expression of dARC1 in the EV-producing cells leads to the increased expression of the protein dARC1 protein and mRNAs in the recipient cells.

---

## [Decision Letter]

**Decision letter after peer review:**

Thank you for submitting your article "An extracellular vesicle targeting ligand that binds to Arc proteins and facilitates Arc transport in vivo" for consideration by *eLife*. Your article has been reviewed by 3 peer reviewers, and the evaluation has been overseen by a Reviewing Editor and K VijayRaghavan as the Senior Editor. The following individual involved in the review of your submission has agreed to reveal their identity: Jason D Shepherd (Reviewer #1).

Essential revisions:

Extracellular vesicles (EVs) are emerging as important mediators of cell-to-cell signaling. In this paper the authors aim to demonstrate that Stranded at second (Sas), a *Drosophila* cell surface protein, binds to dArc1 and Ptp10D to mediate intercellular transport of dArc1 via EVs. dArc1 protein has been shown to form virus-like capsids that carry dArc1 mRNA from neurons to muscle, but little is known about this new intercellular communication pathway. Similarly, not much is known generally about how EVs are targeted to specific cell types, or how specific EV cargo can be delivered. Thus, this work is of interest to cell biologists and neuroscientists. However, the jumbled description of the results and lack of rigor in some experiments diminish the impact and interpretability of the conclusions. Moreover, almost all experiments rely on gain-of-function and over-expression of Sas, thus the relevance to normal physiological signaling is unclear. There are other concerns detailed below that need to be addressed before acceptance. The authors are urged to address concerns raised that can be dealt with using *Drosophila*. The manuscript should also be toned to make it more '*Drosophila*-centric' as generalizations will require additional experiments ( such as with mammalian cell culture). While re-writing, a substantial reduction in the introduction is also suggested. In particular, there is no need to summarise the results in this section in such detail.

We detail below a composite summary of the major points raised by the reviewers for your attention,

The authors' EV isolation protocol is generally not a preferred procedure because it precipitates proteinaceous non-vesicular material, resulting in a mixed preparation of vesicles and protein aggregates. While the authors point to Skotvoll et al., 2019, this paper compares only one approach (ultracentrifugation) to the EV isolation kit. While there is an overlap in the protein components of these approaches in their report, there are considerable differences in the identified proteins. The consensus in the field is that these kits fall under what MISEV2018 Guidelines describe as "high recovery, low specificity" methods, and very likely, these samples have a greater incidence of non-vesicular materials. In this case, the authors' samples are more "secretome-like" and this is observed in their EM of S2 EVs (Figure S2 a,b,d). Using a more robust and specific EV isolation approach, such as a combination of filtration or ultracentrifugation, followed by density gradient fractionation would absolutely improve the quality of their manuscript by providing more convincing evidence that the proposed cargoes are indeed trafficked within EVs. The authors should provide further experimental validation of Sas's EV enrichment, isolating EVs from cultured cells using a more accurate approach.

The description of the results omits some data in the figures and is not in a logical order. This makes the manuscript hard to read and follow. There is also a lack of rigor and quantification in some experiments. Specifically:

1. Figure 1

Lack of quantification: 1 image is just not enough.

Calculate the ratio lysate/EV, also with EVi as a marker for EVs. Only the lysate is shown. EVs should also!

The authors claim (line 153) that most of FL Sas are in EVs? ` It looks like 60%, but it depends on the loading. Please quantify.

Further:

In Figure 1b, the authors suggest that full-length Sas is a predominant EV component, but not Sas-short. The figure would be strengthened by including data describing the in vivo distribution of the V5-tagged short Sas isoform in conjunction with mCD8-GFP. One would expect the Sas-short isoform to be less broadly distributed compared to Sas-FL.

In Supplemental Figure 1, the full blots should be shown for panel A. Why are the control β-Tub bands shown side-by-side, but not the bands for Sas-FL and Sas-short? Also, the authors could include protein samples from wild-type and Sas^-1^5 embryos to enhance the validation of their antibody reagents.

2. Figure 2: Nice IEM and tomography, but many details are needed. Could more IEM profiles (more examples) be provided to be sure of `BG and specificity (in a gallery). Are the capsid reconstitution with purified dARC1 and 2 performed in the presence of darc1 rRNA? Any RNA (figure 2)? Description of dArc1 putative capsids is absent from the Results section (2f,g) until describing Figure 4 data (line 362). Given that there is no immuno-EM labeling of dArc1 protein, it is not clear if Sas and dArc1 are localized to the same EVs. Nor is it clear if the double membrane EVs are actually EVs that contain capsids. Overall, the EM data lacks quantification. How many EVs on average show Sas labeling? How many EVs have double membranes? The immuno-EM data would be strengthened by adding quantifications in the form of distance measurements of gold particles of Anti-V5 and Anti-Sas in relation to EV membranes. The dense protein staining surrounding EVs seems unusual, is this due to an artifact of the purification? EV kits are generally non-specific and isolate non-EV membranes, Corroboration using ultracentrifugation or size exclusion chromatography methods would be beneficial. SAS-FL overexpression results in more EVs, which confounds subsequent experiments suggesting that Sas targets EVs to specific cell types/regions.

3. Figure 3: There are no data showing the expression of Sas in SG cells using the GAL4 lines. Is this expression restricted to just SG cells? (The results jump from a-b to f-g. c-e are out of order. It will be easier if this is ordered). The quantification in g should be broken into two and paired with the actual data (c-e, and f). It is not clear how the quantification in g was performed. How many western blots (WBs) were analyzed? There seems to be a bubble in the first lane of f, which would preclude quantification. Why is d not quantified, and there seems to be an overall increase in background staining in e that is not specific to discs? The source data files are not labeled and these data should be incorporated into annotated supplemental figures. Is transfer in a-b due to Ptp10D? How many WBs were quantified in g? What is the specificity for full-length Sas? The expression of short Sas should not lead to its incorporation in EVs and their overnight addition should not lead to the same effect (Figure 3). This should be better investigated as short Sas is a good control for full-length Sas.

In the salivary gland (SG) expression data presented in Figure 3a-b, the authors should have included V5-Sas-short as a negative control for cross-tissue transport. Similarly, in Figure 3c-e, EVs isolated from Sas-Short transfected cells should have been included as a control in the overnight EV-imaginal disc incubation assays. Does over-expression of Numb on its own lead to increased uptake of Sas-FL EVs? Also, does the need for Ptp10D and Numb over-expression for efficient capture of Sas-FL EVs in Figure 3c-e raise doubts about the data presented in Figure 3a-b, in which the transfer from SGs seems to be efficient without the need for Ptp10D or Numb over-expression?

Also

a: Are wing discs the only recipient tissues upon S2 cells EVs exposure? The tissues in Figure 5 are the trachea and guts. Why switch?

b: What about Ptp10D mutant disc (as in section 2)?

Better quantification? Number of discs etc.

4. Figure 4: C and d, IP data has no inputs for IPs, no sizing markers, and no IgG controls for antibody specificity. These data would also be more convincing if done with FL Sas and included co-Ips from cell lysates.

The IPs presented in Figure 4C with myc-tagged constructs are not controlled properly. Western blots of protein extracts prior to IPs should be shown and a control antibody should be used to show equal loading in the WB for either dArc1-Myc or Myc-rArc-FL expressing cells. Showing the IPs one on top of each other is very confusing and WBs are critical to showing that the expected proteins are expressed in the transfected cells, even if their IP signature is negative.

5. In general, the WBs in the figures show very white backgrounds with high contrast. Please check if any contrast enhancement (or other such changes) were made and if so, they need to be documented and justified. Else, please revert to raw images of blots. Total protein controls are also missing.

6. Figure 5: Ashley et al. (Cell 2018) showed that dArc1 mRNA transfer required the 3'UTR so it is puzzling that the authors used heterologous UTRs. The results using FISH on endogenous dArc1 mRNA are dramatic. The authors should show definitively that their probe does not pick up over-expressed dArc1. In general, a better quantitative analysis should be provided. For instance, there is no quantitative data for Figure 5, but this point should be examined in other figures as well. The dAC1 increased expression in the target cells upon dARC1 increased production in SG(Figure 5) becomes an important part of the paper (and the model) but is not investigated! How does it work? Does the delivery of darc1 mRNAs packaged in capsids simply lead to more dARC1 translation? Is it proportional?

Or, is there also stimulation of darc1 transcription? Is there also an increase in the mRNA level (we cannot see the SG control of 5o (sage>+) supporting the authors' claim on line 562!).

Also:

a: What is the dARC1 LOF mutant phenotype in SG and trachea?

b: Do EV produced upon FL Sas v5 expression in an SG not produce dARC1 in the recipient trachea?

c: Does this also depend on the same synergetic presence of Ptp10D and Numb in the trachea? Ptp10D LOF?

The authors' data suggest that dArc1 protein and mRNA are transferred by Sas-FL expressing EVs to Ptp10D enriched tissues in fly embryos. Could the authors conduct immuno-EM or nano-flow cytometry studies to assess the general proportion of Sas and dArc1 co-expression in purified EVs? If the general model is accurate, one would expect a higher degree of co-occurrence compared to control EV cargos. Additionally, evaluating the dArc1 protein and mRNA transfer properties in different genetic backgrounds (i.e. Sas-FL and Sas-Short in Ptp10D over-expressed or in Ptp10D null embryos) would significantly enhance the paper.

Also in Figure 5, the results suggesting that transgenic over-expression of dArc1 leads to a strongly increased expression of endogenous dArc1 should be validated using a complementary strategy (e.g. RT-qPCR on dissected tissue specimens, RNA-FISH with probes specific to the endogenous mRNA).

- For all the tissue imaging studies, including those in Figures 5-6,, the authors should provide an indication of how many specimens were analyzed in total, presuming the samples shown in the figures were representative of observations made across multiple tissue specimens co-labeled in the same experiment.

7. Figure 6: what is the red circle (and blue)?

What is the black circle, an SG cell?

8. Most (all) experiments are performed with over-expression of FL Sas or ICD. Does endogenous Sas bind endogenous Ptp10D and dARC1? ICDs? Also full-length APP? Many of the conclusions would be strengthened by the loss of function experiments, especially showing a requirement for Sas in dArc1 transfer.

9. Use of more extensive fly genetics using specific Ptp10D LOF in wing discs and trachea (to show the converse of the GOF). Does Ptp10D acts as the MAIN receptor to FL Sas? Numb LOF, a combination of LOF and GOF? Does Ptp10D GOF compensate for Numb and vice versa?

– The introduction section would benefit from some sentence restructuring, synthesizing major ideas into a more coherent train of thought. For instance, very short, matter-of-fact statements such as "EV cargoes are biomarkers for specific diseases" are not helpful, especially since no specific examples of diseases are provided. Also, more primary sources of literature should be cited, as many statements are lacking supporting evidence, particularly those describing EV biology and characteristics.

– While the authors use "comparatively" to describe the current knowledge of EV-biogenesis pathways as "well understood," this is not an appropriate description and can be misleading. EV-biogenesis pathways are only beginning to be unraveled.

– When describing their major finding (Lee et al., 2013) that Ptp10D is a binding partner for Sas, The authors should stress that this was done in a previous study. Simply adding a world like "previously" will make this clear.

– The paragraph from lines 85-90 stating that 'Sas localizes to EVs, as demonstrated…' appears to be out of context and is more appropriate for the 'Results' section.

– The authors should mention what device they used for NTA, not just provide the name of the outsourcing company. From their description, it appears they used NanoSight, but don't provide acquisition parameters.

– In Figure 2 f,g the authors provide EM images of capsid-like structures for dArc1 and dArc2, however, they do not address these until much later in the manuscript, when they describe how these capsids were generated. The authors should consider reorganizing their figures to improve the flow of their narration and not group them into categories, as this makes the manuscript more difficult to follow.

– There may be referencing errors in the manuscript when mentioning figures, for instance when they describe the detection of V5-mCD8ECD-SasTM-ICD by western blotting, they reference Figure 4d, but it appears they actually mean Figure 4c. Another such error appears in the text describing Figure 5n, which depicts *Drosophila* embryos co-expressing SasFL and dArc1, and which authors describe as Figure 5m. I recommend that they carefully double-check the manuscript for any such errors, which are confusing for the reader.

– When appropriate the authors should add statistical analysis to their quantitative data, for example, Figure 3g.

---

## [Author Response]

Essential revisions:The authors' EV isolation protocol is generally not a preferred procedure because it precipitates proteinaceous non-vesicular material, resulting in a mixed preparation of vesicles and protein aggregates. While the authors point to Skotvoll et al., 2019, this paper compares only one approach (ultracentrifugation) to the EV isolation kit. While there is an overlap in the protein components of these approaches in their report, there are considerable differences in the identified proteins. The consensus in the field is that these kits fall under what MISEV2018 Guidelines describe as "high recovery, low specificity" methods, and very likely, these samples have a greater incidence of non-vesicular materials. In this case, the authors' samples are more "secretome-like" and this is observed in their EM of S2 EVs (Figure S2 a,b,d). Using a more robust and specific EV isolation approach, such as a combination of filtration or ultracentrifugation, followed by density gradient fractionation would absolutely improve the quality of their manuscript by providing more convincing evidence that the proposed cargoes are indeed trafficked within EVs. The authors should provide further experimental validation of Sas's EV enrichment, isolating EVs from cultured cells using a more accurate approach.

We have done this. We isolated EVs from S2 cells expressing V5-tagged Sas^FL^ using a standard ultracentrifugation protocol (Thery et al., 2006). We observed that Sas^FL^ was also present in EVs isolated in this manner (new Supp. Figure 1). The presence of dArc1 and *dArc1* mRNA in EVs from S2 cells is not in question, since this has been published by several other groups.

Figure 1Lack of quantification: 1 image is just not enoughCalculate the ratio lysate/EV, also with EVi as a marker for EVs. Only the lysate is shown. EVs should also!The authors claim (line 153) that most of FL Sas are in EVs? ` It looks like 60%, but it depends on the loading. Please quantify.

We quantitated the relative amounts of Sas^FL^ and Sas^short^ in EVs and lysates, and these data are now reported in the paper. On average, 46% of Sas^FL^ and 10% of Sas^short^ are in EVs. We also now use Wg as an EV marker instead of Evi, because Wg is almost absent from lysates. Wg was originally shown to be a marker for EVs from S2 cells by Koles et al., 2012. We observed that Wg was present in EVs isolated with the kit and by ultracentrifugation.

In Figure 1b, the authors suggest that full-length Sas is a predominant EV component, but not Sas-short. The figure would be strengthened by including data describing the in vivo distribution of the V5-tagged short Sas isoform in conjunction with mCD8-GFP. One would expect the Sas-short isoform to be less broadly distributed compared to Sas-FL.

We have done this, and the data are in the new Figure 1—figure supplement 1. The data show that Sas^short^ fails to move into the extracellular matrix when it is expressed in Ap neurons.

The authors claim (line 153) that most of FL Sas are in EVs? ` It looks like 60%, but it depends on the loading. Please quantify. In Supplemental Figure 1, the full blots should be shown for panel A. Why are the control β-Tub bands shown side-by-side, but not the bands for Sas-FL and Sas-short? Also, the authors could include protein samples from wild-type and Sas^-1^5 embryos to enhance the validation of their antibody reagents.

We have done the quantification. On average, 46% of Sas^FL^ is in EVs, and 10% of Sas^short^. In the blot shown in Figure 1, 60% of Sas^FL^ is in EVs, and 10% of Sas^short^. We replaced the blot of Figure 1—figure supplement 1 to show the two isoforms side-by-side. It is not possible to do western blots on *sas* mutant embryos because it is a lethal mutant, so there is no way to generate a pure population of embryos.

In Supplemental Figure 1, the full blots should be shown for panel A. Why are the control β-Tub bands shown side-by-side, but not the bands for Sas-FL and Sas-short?

We have replaced this blot. We now show 3 sections of adjacent lanes of the same gel.

Figure 3: There are no data showing the expression of Sas in SG cells using the GAL4 lines. Is this expression restricted to just SG cells?

We now include data (Figure 3—figure supplement 1) showing that Sage-GAL4 expresses only in SGs.

Figure 3: It is not clear how the quantification in g was performed. How many western blots (WBs) were analyzed? There seems to be a bubble in the first lane of f, which would preclude quantification. Why is d not quantified, and there seems to be an overall increase in background staining in e that is not specific to discs? The source data files are not labeled and these data should be incorporated into annotated supplemental figures. Is transfer in a-b due to Ptp10D? How many WBs were quantified in g? What is the specificity for full-length Sas? The expression of short Sas should not lead to its incorporation in EVs and their overnight addition should not lead to the same effect (Figure 3). This should be better investigated as short Sas is a good control for full-length Sas.

Six WBs were used for quantification, and five discs for each genotype. Transfer in Figure 3b may involve Ptp10D, since Ptp10D is expressed at low levels in wt discs (Figure 3—figure supplement 1). Transfer does not require Ptp10D, however, since there is transfer to untransfected S2 cells, which lack Ptp10D. Our earlier data (Lee et al., 2013) shows that Sas also has additional, unidentified binding partners. There would be no point in adding supernatant from S2 cells transfected with Sas^short^, since there is almost no Sas^short^ in such supernatants.

Figure 3, contd. In the salivary gland (SG) expression data presented in Figure 3a-b, the authors should have included V5-Sas-short as a negative control for cross-tissue transport. Similarly, in Figure 3c-e, EVs isolated from Sas-Short transfected cells should have been included as a control in the overnight EV-imaginal disc incubation assays. Does over-expression of Numb on its own lead to increased uptake of Sas-FL EVs? Also, does the need for Ptp10D and Numb over-expression for efficient capture of Sas-FL EVs in Figure 3c-e raise doubts about the data presented in Figure 3a-b, in which the transfer from SGs seems to be efficient without the need for Ptp10D or Numb over-expression?Alsoa: Are wing discs the only recipient tissues upon S2 cells EVs exposure? The tissues in Figure 5 are the trachea and guts. Why switch?b: What about Ptp10D mutant disc (as in section 2)?Better quantification? Number of discs etc

We have added an experiment showing that when Sas^short^ is expressed in SGs, it does not move to discs (new Figure 3c). The experiment of adding supernatant from Sas^short^-expressing cells is not worth doing, for the reasons explained above. We did some experiments with Numb alone, and it had no effect, so we did not include this. The transfer experiments don’t raise questions about Figure 3b for the reasons explained above. Ptp10D stimulates transfer but is not required. There is low-level Ptp10D expression in discs, and Sas has other binding partners. Transfer is much more efficient when Sas^FL^ is expressed in SGs in vivo than when supernatant is added to discs. We used discs because they can be easily dissected away from larvae. Tracheae are embedded in the body wall, and the larval gut is huge and hard to visualize. In embryos, tracheae and gut can be visualized in whole-mounts, but that is not possible in 3^rd^ instar larvae. It is not worth doing *Ptp10D* LOF experiments. These would be time-consuming, requiring months to combine the LOF allele with drivers and reporters, and would likely not produce clear results, since Sas has other binding partners. We now report the number of discs measured (5 per condition).

Figure 4: C and d, IP data has no inputs for IPs, no sizing markers, and no IgG controls for antibody specificity. These data would also be more convincing if done with FL Sas and included co-Ips from cell lysates.The IPs presented in Figure 4C with myc-tagged constructs are not controlled properly. Western blots of protein extracts prior to IPs should be shown and a control antibody should be used to show equal loading in the WB for either dArc1-Myc or Myc-rArc-FL expressing cells. Showing the IPs one on top of each other is very confusing and WBs are critical to showing that the expected proteins are expressed in the transfected cells, even if their IP signature is negative.5. In general, the WBs in the figures show very white backgrounds with high contrast. Please check if any contrast enhancement (or other such changes) were made and if so, they need to be documented and justified. Else, please revert to raw images of blots. Total protein controls are also missing.

We have replaced the WBs with new blots showing lysate blotted with each antibody. The CD8 chimera is better than Sas^FL^ because it isolates the binding region to the ICD and also produces cleaner blots than Sas^FL^, being a much shorter protein.

Figure 5: Ashley et al. (Cell 2018) showed that dArc1 mRNA transfer required the 3'UTR so it is puzzling that the authors used heterologous UTRs. The results using FISH on endogenous dArc1 mRNA are dramatic. The authors should show definitively that their probe does not pick up over-expressed dArc1. In general, a better quantitative analysis should be provided. For instance, there is no quantitative data for Figure 5, but this point should be examined in other figures as well. The dAC1 increased expression in the target cells upon dARC1 increased production in SG(Figure 5) becomes an important part of the paper (and the model) but is not investigated! How does it work? Does the delivery of darc1 mRNAs packaged in capsids simply lead to more dARC1 translation? Is it proportional?Or, is there also stimulation of darc1 transcription? Is there also an increase in the mRNA level (we cannot see the SG control of 5o (sage>+) supporting the authors' claim on line 562!).Also:a: What is the dARC1 LOF mutant phenotype in SG and trachea?b: Do EV produced upon FL Sas v5 expression in an SG not produce dARC1 in the recipient trachea?c: Does this also depend on the same synergetic presence of Ptp10D and Numb in the trachea? Ptp10D LOF?– For all the tissue imaging studies, including those in Figures 5-6,, the authors should provide an indication of how many specimens were analyzed in total, presuming the samples shown in the figures were representative of observations made across multiple tissue specimens co-labeled in the same experiment.

We used the dArc1 ORF construct so that we could distinguish the mRNA from endogenous *dArc1* mRNA, and because it was reported to express well. The probe we used cannot detect the transgene RNA because that does not contain any of the *dArc1* 3’ UTR sequences in the probe. We now include new FISH experiments with an SV40 3’ UTR probe, which picks up only the mRNA from the transgene, and this confirms the results of Ashley et al., showing that the transgene mRNA cannot move away from the SGs because it is not loaded into capsids. Regarding the other criticisms of Figure 5: (1) it is beyond the scope of this paper to define the mechanism by which dArc1 protein increases production of the endogenous *dArc1* mRNA. It could be due to de novo transcription or to mRNA stabilization. (2) there are no embryonic *dArc1* LOF phenotypes that have been detected. (3) It would be difficult and time-consuming to incorporate *Ptp10D* LOF mutations into a genetic background that already contains several transgenes and a driver. In any case, it is likely that Sas has other binding partners, as demonstrated by our original paper (Lee et al., 2013), so we would not expect that transfer would be eliminated. (3) The FISH experiments are representative of the results from >100 embryos. Such experiments in *Drosophila* are normally not quantitated. (4) We show that translocation of *dArc1* mRNA requires expression of both Sas^FL^ and dArc1. dArc1 capsids are a normal EV component, but our data show that without Sas^FL^ they cannot move to tracheae. It is difficult to determine what percentage of EVs have both Sas^FL^ and dArc1, because in immuno-EM experiments there are always many antigens that do not have gold particles, and in any case we were not able to do immuno-EM with anti-dArc1. (5) We now show data with both *dArc1* 3’ UTR and SV40 3’ UTR probes, showing that the transgenic construct expresses in SGs but its mRNA is not observed outside of the expressing cells.

Responses to the remaining criticisms in the composite review

2. Figure 2: Nice IEM and tomography, but many details are needed. Could more IEM profiles (more examples) be provided to be sure of `BG and specificity (in a gallery). Are the capsid reconstitution with purified dARC1 and 2 performed in the presence of darc1 rRNA? Any RNA (figure 2)? Description of dArc1 putative capsids is absent from the Results section (2f,g) until describing Figure 4 data (line 362). Given that there is no immuno-EM labeling of dArc1 protein, it is not clear if Sas and dArc1 are localized to the same EVs. Nor is it clear if the double membrane EVs are actually EVs that contain capsids. Overall, the EM data lacks quantification. How many EVs on average show Sas labeling? How many EVs have double membranes? The immuno-EM data would be strengthened by adding quantifications in the form of distance measurements of gold particles of Anti-V5 and Anti-Sas in relation to EV membranes. The dense protein staining surrounding EVs seems unusual, is this due to an artifact of the purification? EV kits are generally non-specific and isolate non-EV membranes, Corroboration using ultracentrifugation or size exclusion chromatography methods would be beneficial. SAS-FL overexpression results in more EVs, which confounds subsequent experiments suggesting that Sas targets EVs to specific cell types/regions.

1) In immuno-electron microscopy of embedded material, regardless of the methodology used, antibodies do not penetrate beyond the surfaces of the physical section. Because of this, only epitopes that are exposed at the section surfaces are available to and labeled by the antibodies. Thus, if a portion of an EV is at the section surface it can be labeled. However, if the entirety of the EV is contained within the volume of the section and no part of it is exposed at the surface, it is not accessible to the antibodies and will not be labeled (despite the fact that densities corresponding to the structure can still be seen). Because of this, overview images and/or galleries may show both labeled and unlabeled structures, but those that are unlabeled cannot be interpreted or quantified as being negative for the specific probe. (2) The capsids made from dArc1 and dArc2 made in *E. coli* do not have added RNA but probably incorporate *E. coli* RNA. In any case, we have removed these data, as they repeat published data on the capsids and don’t add value to the paper. (3) We have not been able to obtain immuno-EM labeling for dArc1. The antibody is of low quality and we had only a very small quantity of it. However, it is already known from Ashley et al., 2018 and others that EVs from untransfected S2 cells, which express Sas^FL^ only at very low levels, often contain dArc1 capsids and *dArc1* mRNA. (4) There is no point in doing distance measurements since the Sas ECD is a flexible structure that would not have a defined length. In fact, the observed distance is highly variable, probably because the Sas ECD, which is composed of a chain of domains separated by linkers, is likely to be flexible and able to adopt many different conformations. (5) It is true that Sas^FL^ expression modestly increases EV production, but the effects observed in Figure 5 are all-or-nothing, so this cannot account for translocation of *dArc1* mRNA to other cells. 6) We have purified EVs using ultracentrifugation, and these data are reported in Figure 1. We did not think it worthwhile to do a new series of EM experiments just to visualize S2 EVs purified in this manner, since this has already been published.

The authors' data suggest that dArc1 protein and mRNA are transferred by Sas-FL expressing EVs to Ptp10D enriched tissues in fly embryos. Could the authors conduct immuno-EM or nano-flow cytometry studies to assess the general proportion of Sas and dArc1 co-expression in purified EVs? If the general model is accurate, one would expect a higher degree of co-occurrence compared to control EV cargos. Additionally, evaluating the dArc1 protein and mRNA transfer properties in different genetic backgrounds (i.e. Sas-FL and Sas-Short in Ptp10D over-expressed or in Ptp10D null embryos) would significantly enhance the paper.Also in Figure 5, the results suggesting that transgenic over-expression of dArc1 leads to a strongly increased expression of endogenous dArc1 should be validated using a complementary strategy (e.g. RT-qPCR on dissected tissue specimens, RNA-FISH with probes specific to the endogenous mRNA).

See above. We were not able to do immuno-EM with antibody against dArc1 and in any case this technique is not quantitative. One could not draw any conclusions from counting EVs that appeared to be labeled for both antigens vs. only one (see discussion above). We did do FISH with probes specific for the endogenous mRNA, as discussed above.

Most (all) experiments are performed with over-expression of FL Sas or ICD. Does endogenous Sas bind endogenous Ptp10D and dARC1? ICDs? Also full-length APP? Many of the conclusions would be strengthened by the loss of function experiments, especially showing a requirement for Sas in dArc1 transfer.8. Use of more extensive fly genetics using specific Ptp10D LOF in wing discs and trachea (to show the converse of the GOF). Does Ptp10D acts as the MAIN receptor to FL Sas? Numb LOF, a combination of LOF and GOF? Does Ptp10D GOF compensate for Numb and vice versa?

1) Ectopic expression of Sas and dArc1 is the only way to evaluate transfer from one cell type to another, since both genes are broadly expressed. (2) There is no way to determine if endogenous Sas binds to dArc1. Endogenous Sas, however, is found in cells adjacent to those expressing Ptp10D (Lee et al., 2014; Yamamoto et al. 2017), suggesting that the two proteins might form a complex. (3) There would be no reason to express full-length APP in S2s, since we have already demonstrated that dArc1 and Arc binding can be localized to the APP ICD using the CD8 chimera. In any case, the rest of APP is on the outside of the cell/EV and therefore not accessible to Arcs. (4) One cannot show a requirement for Sas in dArc1 transfer. This would be possible to test if it was observed that *dArc1* mRNA moved to other cells in the absence of Sas^FL^ coexpression in SGs (since one could then remove Sas and ask if transfer still occurred), but since *dArc1* mRNA remains in the SGs unless Sas^FL^ is coexpressed this experiment cannot be done. (5) It would be very time-consuming to introduce *Ptp10D* LOF mutations into genetic backgrounds containing 3 or more transgenes (as in Figure 5). (6) Numb is an early embryonic lethal, so *numb* mutant embryos cannot be examined.